# Active and dynamic mitochondrial S-depalmitoylation revealed by targeted fluorescent probes

Rahul S. Kathayat[1], Yang Cao[1], Pablo D. Elvira[1], Patrick A. Sandoz[2], María-Eugenia Zaballa[2], Maya Z. Springer[3,4], Lauren E. Drake[4], Kay F. Macleod[3,4], F. Gisou van der Goot [2] & Bryan C. Dickinson [1,4]

The reversible modification of cysteine residues by thioester formation with palmitate (S-palmitoylation) is an abundant lipid post-translational modification (PTM) in mammalian systems. S-palmitoylation has been observed on mitochondrial proteins, providing an intriguing potential connection between metabolic lipids and mitochondrial regulation. However, it is unknown whether and/or how mitochondrial S-palmitoylation is regulated. Here we report the development of mitoDPPs, targeted fluorescent probes that measure the activity levels of "erasers" of S-palmitoylation, acyl-protein thioesterases (APTs), within mitochondria of live cells. Using mitoDPPs, we discover active S-depalmitoylation in mitochondria, in part mediated by APT1, an S-depalmitoylase previously thought to reside in the cytosol and on the Golgi apparatus. We also find that perturbation of long-chain acyl-CoA cytoplasm and mitochondrial regulatory proteins, respectively, results in selective responses from cytosolic and mitochondrial S-depalmitoylases. Altogether, this work reveals that mitochondrial S-palmitoylation is actively regulated by "eraser" enzymes that respond to alterations in mitochondrial lipid homeostasis.

[1] Department of Chemistry, The University of Chicago, Chicago, IL 60637, USA. [2] Global Health Institute, School of Life Sciences, Ecole Polytechnique Fédérale de Lausanne, Lausanne CH-1015, Switzerland. [3] The Ben May Department for Cancer Research, The University of Chicago, Chicago, IL 60637, USA. [4] The Committee on Cancer Biology, The University of Chicago, Chicago, IL 60637, USA. Correspondence and requests for materials should be addressed to F.G.v.d.G. (email: Gisou.vandergoot@epfl.ch) or to B.C.D. (email: Dickinson@uchicago.edu)

The lipidation of proteins through thioester modification of cysteine residues by palmitate (S-palmitoylation) is an abundant mammalian post translational modification (PTM) present on hundreds of protein targets[1]. Unlike most lipid PTMs, S-palmitoylation is reversible, providing a mechanism to dynamically control the hydrophobicity and cellular localization of target proteins[2–5]. The S-palmitoylation status of a given protein is regulated by, (1) local concentrations of palmitoyl-CoA, the source of the acyl modification, (2) the "writer" enzymes that transfer the lipid moiety from palmitoyl-CoA to a target protein, and (3) cysteine deacylase "eraser" enzymes that remove the modification. In human, the known "writer" proteins are comprised of 23 Asp–His–His–Cys (DHHC)-containing transmembrane protein acyltransferases (DHHC–PATs), which localize to various intracellular compartments, including the endoplasmic reticulum, the Golgi, and the plasma membrane[4,6]. The known "eraser" proteins include Palmitoyl-Protein Thioesterase 1 (PPT1), a lysosomal protein, and the presumed cytosolic proteins acyl-protein thioesterase 1 (APT1; also known as Lysophospholipase 1, LYPLA1) and acyl-protein thioesterase 2 (APT2; also known as Lysophospholipase 2, LYPLA2)[7–11]. In addition, the α/β-Hydrolase domain-containing protein 17 members A, B, and C (ABHD17A/B/C), three other members of the metabolic serine hydrolase (mSH) superfamily, were recently uncovered as depalmitoylases targeting critical proteins including N-Ras and PSD-95[12–14].

To date, most research has focused on the role of protein S-palmitoylation of cytosolic targets or transmembrane proteins of the endomembrane system that are regulated by DHHC–PATs and cytosolic APTs. However, proteomic studies have revealed potentially hundreds of palmitoylated proteins in the mitochondria[15–17]. For example, the mitochondrial proteins methylmalonate semialdehyde dehydrogenase (MMSDH), glutamate dehydrogenase, and carbamoyl-phosphate synthetase 1 (CPS 1) have been shown to be regulated by palmitoylation to control mitochondrial energy levels[18,19]. Critical disease-relevant metabolic proteins, such as the rate-limiting enzyme in ketogenesis, mitochondrial HMG-CoA synthase (HMGCS2), are palmitoylated[20]. The electron-transferring flavoprotein (ETF), sarcosine dehydrogenase, isovaleryl-CoA dehydrogenase, and dimethylglycine dehydrogenase are all palmitoylated, suggesting this lipid modification may also play a role in membrane localization and function of the electron transport chain, the central component of mitochondria. Mitochondrial fission induction by the GTPase Irgm1[21] is controlled through its palmitoylation. Finally, palmitoylation serves as a molecular switch to regulate the trafficking of BCL-2-associated X (BAX) to the mitochondria to induce apoptosis[22].

Intuitively, it makes sense that mitochondria, a primary driver of metabolic regulation and source of long chain acyl-CoA donors[23], would utilize dynamic protein lipidation as a PTM regulatory mechanism. However, the presence of mitochondrial palmitoylated proteins is also surprising, as none of the protein acyltransferases are known to function in the mitochondria[24]. The lack of known mitochondrial acyltransferases, combined with the observation that many of the identified palmitoylated proteins are dehydrogenases, has led to the suggestion that mitochondrial palmitoylation may be non-enzymatic, and perhaps mediated by NAD(H) binding sites on target proteins[15]. Such a mechanism would provide a direct chemical link between energy status and mitochondrial protein lipidation, as the levels of protein palmitoylation would be sensitive to the levels of palmitoyl-CoA, which has long been known to regulate mitochondrial function through a variety of mechanisms[25–27].

Whether the installation of the S-palmitoyl group on a target mitochondrial protein is enzymatic or non-enzymatic, the abundance of the modification will be at least in part determined by its removal. Thioesters are quite stable to base-mediated hydrolysis[28], even at the elevated pH of the mitochondria. Therefore, if mitochondrial S-palmitoylation is indeed a regulatory mechanism, it seems likely that there would be active "eraser" enzymes within this cellular compartment to remove the modification. There are no reports of APT1 or APT2, or the ABHD17A/B/C proteins, functioning in mitochondria. However, unbiased mitochondrial proteomic studies both in mice[29–31] and humans[32,33], each using mitochondria isolated from a variety of tissue types, suggest that APT1, at least, may be present in mitochondria.

In this work, we sought to experimentally test whether there is active S-depalmitoylation in the mitochondria of mammalian cells, whether APT1 is active in the mitochondria, and whether changes to mitochondrial homeostasis result in alterations of mitochondrial S-depalmitoylation activity. To accomplish this, we expanded our recently reported strategy for the creation of fluorescent "depalmitoylation probes" (DPPs)[34], which readout endogenous cysteine deacylation activities in live cells, to generate the first mitochondrial-targeted S-deacylase probes ("mitoDPP-2" and "mitoDPP-3"). After synthesis, we validated that mitoDPPs localize to mitochondria and report on endogenous cysteine S-deacylase activities. We found that inhibition of cellular depalmitoylases blocks the mitochondrial signal from the mitoDPPs, indicating that the S-deacylase signal is due to active enzymatic thioester cleavage in the mitochondria. We discovered that APT1 is responsible for part of the measured mitochondrial S-deacylase signal, revealing a new function for this key S-palmitoylation eraser protein. Through immunostaining and fractionation experiments, we show that APT1 is in fact primarily localized at mitochondria and not the cytosol and the Golgi apparatus, as has been previously reported[11,35,36]. We then discovered that lipid stress causes a response from the mitochondrial S-deacylases. Finally, we discovered that knockdown of ACOT11, a long-chain acyl-CoA thioesterase that regulates mitochondrial lipids, results

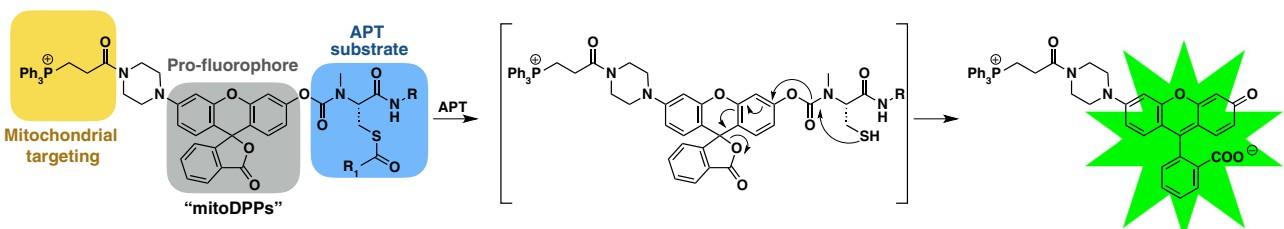

**Fig. 1** Design and activation of mitoDPPs. MitoDPPs localize to the mitochondria via an appended triphenylphosphonium group, which delivers the probe to the mitochondria based on the electrochemical potential. The carbamate-linked APT substrate forces the fluorophore into the lactone-closed, non-fluorescent form. Cleavage of the thioester by reaction with an APT releases the thiol, which rapidly cleaves the carbamate, generating a fluorescent product

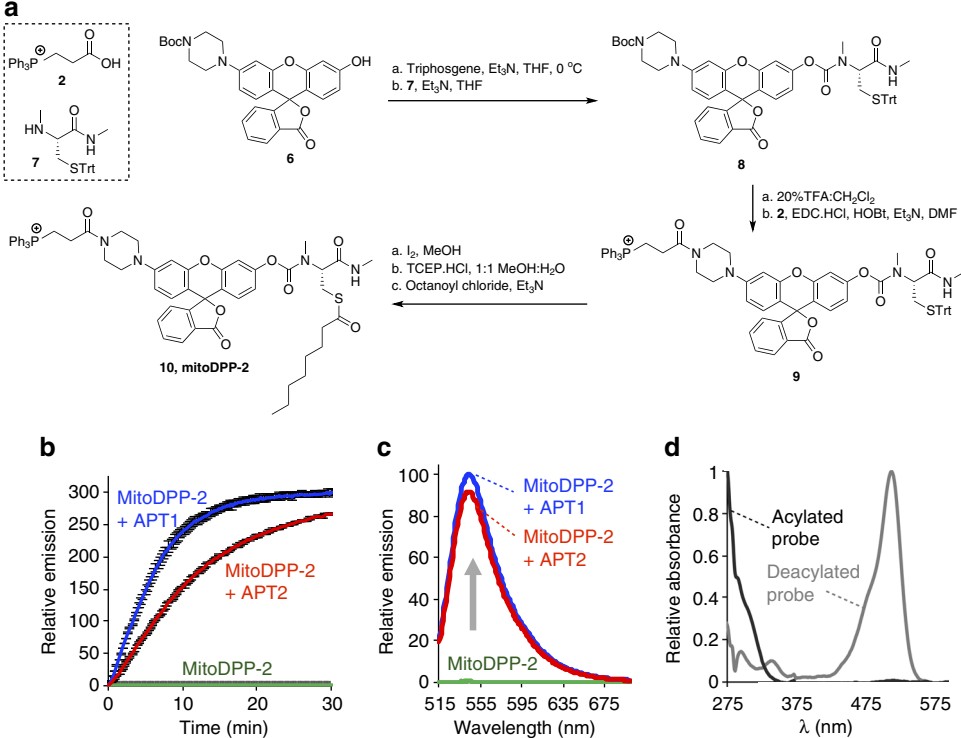

**Fig. 2** Synthesis and in vitro activity of mitoDPP-2. **a** Synthetic scheme for mitoDPP-2. **b** In vitro fluorescence assay of mitoDPP-2 (1 μM) in HEPES (20 mM, pH 7.4, 150 mM NaCl, 0.1% Triton X-100) with either 50 nM purified APT1 or APT2 ($\lambda_{ex}$ 490/20 nm, $\lambda_{em}$ 545/20 nm). Error bars are ± s.e.m. ($n = 3$). **c** Fluorescence emission spectra at 30 min from probes as treated in (**b**). **d** UV-vis spectra of 25 μM mitoDPP-2 (black; normalized at 275 nm) and deprotected fluorophore product (gray; normalized at 513 nm) in HEPES (20 mM, pH 7.4, 150 mM NaCl, 0.1% Triton X-100). MitoDPP-2 shows UV-vis absorbance at 300 nm with extinction coefficient $8.7 \times 10^3 \, M^{-1} \, cm^{-1}$. The deprotected fluorophore product shows a major UV-Vis absorbance peak at 513 nm with extinction coefficient $11.8 \times 10^3 \, M^{-1} \, cm^{-1}$

in increased mitochondrial *S*-deacylation activity without affecting cytosolic *S*-deacylation levels, while knockdown of the cytosolic lipid regulatory ACOT protein, ACOT1, had the opposite effect. Altogether, this work demonstrates that mitochondria contain APTs, whose activity is actively and dynamically regulated, and that mitoDPPs are valuable new tools to study the regulation of *S*-deacylation specifically in this compartment.

## Results

**Design and synthesis of mitoDPP-2**. To create a mitochondrial-targeted *S*-deacylation probe, we envisioned adopting our recently developed DPP strategy[34,37,38] by appending a mitochondrial delivery group to the probe scaffold. For delivery, we sought to deploy the lipophilic cation triphenylphosphonium (TPP) moiety, which can shuttle cargo to the mitochondria based on the electrochemical gradient[39–41]. The first-generation DPPs utilized a rhodol scaffold, which affords several synthetically-tractable positions to install additional functional groups. We postulated that a mitochondrial-targeted rhodol scaffold that uses a piperazine linker to append a TPP-targeting group, which has been successfully deployed to deliver $H_2O_2$ probes to the mitochondria[42–44], could be adopted for the DPP strategy. Therefore, we designed mitoDPP-2, which utilizes the APT-sensing group from the original DPPs, the octanyl cysteine acyl substrate that we originally found to maintain APT activity but which enhances cell uptake, and a piperazine linker to tether a TPP-targeting group to the probe (Fig. 1). Synthesis of mitoDPP-2 proceeded smoothly over three steps (Fig. 2a). Detailed synthetic procedures and chemical characterization is outlined in Supplementary Methods.

**In vitro characterization of mitoDPP-2**. Upon synthesis, we tested whether mitoDPP-2 responds to recombinant human APTs in vitro with enhanced fluorescence. 1 μM of mitoDPP-2 displays very low fluorescence in buffer ($\lambda_{em} = 545$ nm), but has a dramatic increase in fluorescence upon incubation with 50 nM recombinantly expressed and purified human APT1 or APT2 (Fig. 2b, c, Supplementary Table 1, Supplementary Fig. 1). The fluorescence turn-on response correlates with a concomitant absorbance enhancement (Fig. 2d), indicating that the primary mechanism modulating the fluorescence response of the mitoDPPs is through lactone opening and a shift in the equilibrium from the xanthene-like structure to the fully-conjugated lactone-open structure. Kinetic analysis of the fluorescent response of mitoDPP-2 indicates it responds rapidly to both APT1 and APT2, with slightly more APT1 response (Fig. 2b). Furthermore, in vitro kinetic assays in buffer conditions that mirror the mitochondrial environment confirms mitoDPP-2 still functions without any significant hydrolysis (Supplementary Fig. 2). Because mitoDPP-2 showed a 300-fold enhancement in fluorescence emission after 30 min, we decided to pursue live cell imaging experiments.

**MitoDPP-2 reveals active mitochondrial *S*-deacylation**. To assess the localization of mitoDPP-2 in live cells, we used confocal fluorescence microscopy to image human cell lines loaded with both mitoDPP-2 and organelle stains. First, we determined localization in HEK293T cells, which we loaded with mitoDPP-2, the mitochondrial stain MitoTracker, and the nuclear stain Hoechst 33342. As shown in Fig. 3a (Supplementary Fig. 3), we

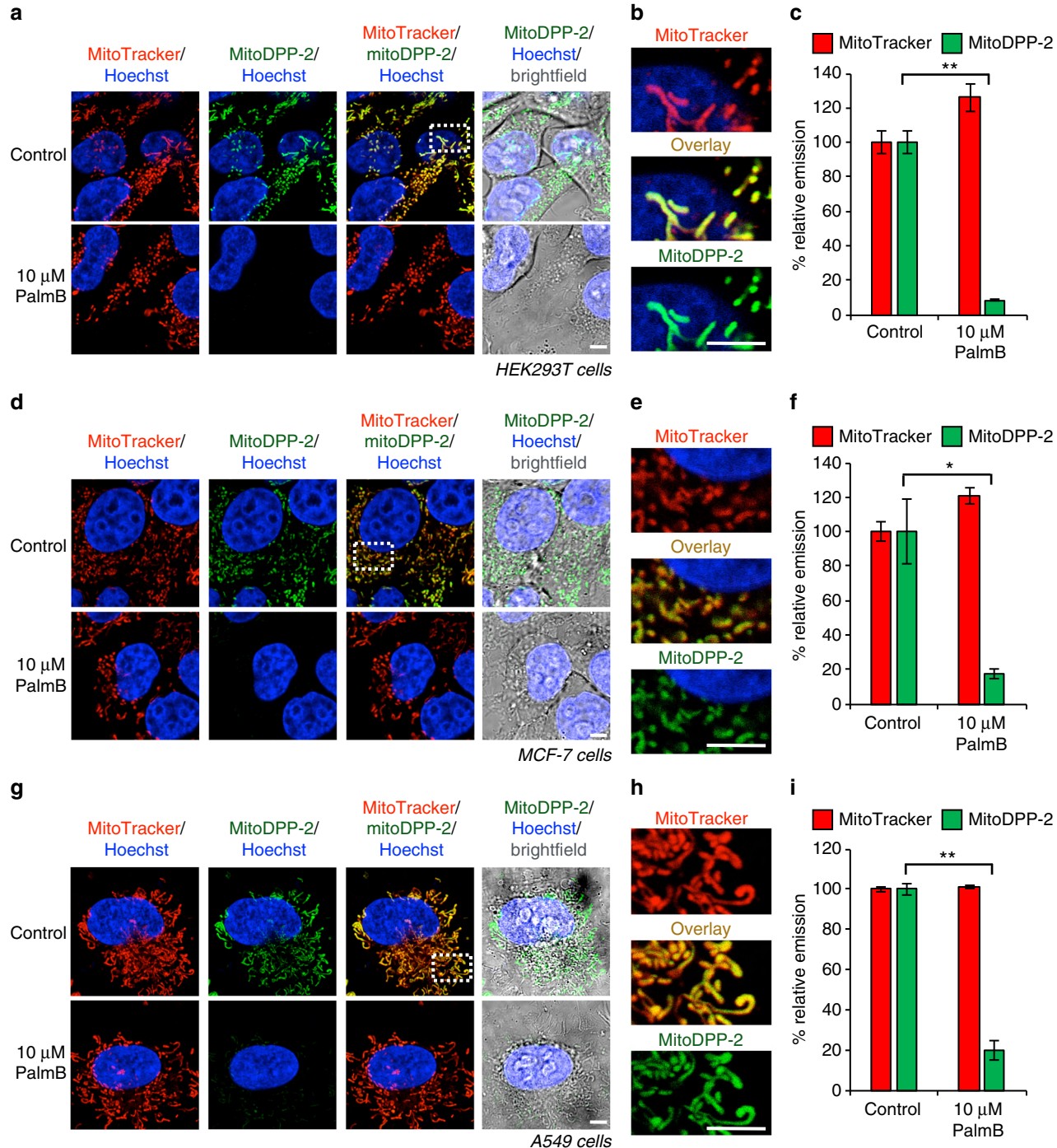

**Fig. 3** Localization and PalmB inhibition of mitoDPP-2 in multiple cell lines. **a** HEK293T cells treated for 30 min with 1 µM Hoechst 33342, 100 nM MitoTracker Deep Red, and either DMSO or 10 µM PalmB, washed, loaded with 250 nM mitoDPP-2 for 10 min, and then analyzed by confocal fluorescence microscopy. **b** Enlarged portion of image indicated by dotted white line in (**a**) showing colocalization of mitoDPP-2 with MitoTracker. **c** Quantification of the relative fluorescence intensity from mitoDPP-2 and MitoTracker in either control or PalmB-treated cells from (**a**). **d** MCF-7 cells treated and analyzed identically to conditions in (**a**). **e** Enlarged portion of image indicated by dotted white line in (**d**). **f** Quantification of the relative fluorescence intensity from mitoDPP-2 and MitoTracker in either control or PalmB-treated cells from (**d**). **g** A549 cells treated and analyzed identically to conditions in (**a**). **h** Enlarged portion of image indicated by dotted white line in (**g**). **i** Quantification of the relative fluorescence intensity from mitoDPP-2 and MitoTracker in either control or PalmB-treated cells from (**g**). For all imaging, 5 µm scale bar shown. For all plots, statistical analyses performed with a two-tailed Student's $t$-test with unequal variance, $*P$ value $< 0.05$; $**P$ value $< 0.005$, $n = 3$, error bars are $\pm$ s.e.m

observed robust, punctate fluorescent signal from the mitoDPP-2 channel. The signal from mitoDPP-2 colocalizes with Mito-Tracker (Fig. 3b), indicating that mitoDPP-2 is successfully delivered to the mitochondria as designed. To assess whether

mitoDPP-2 also accumulates in the mitochondria in other cell lines, we repeated the confocal imaging experiments in MCF-7 and A549 cells. Similar to the results in HEK293T cells, mitoDPP-2 shows robust punctate signal in each cell line, which colocalizes

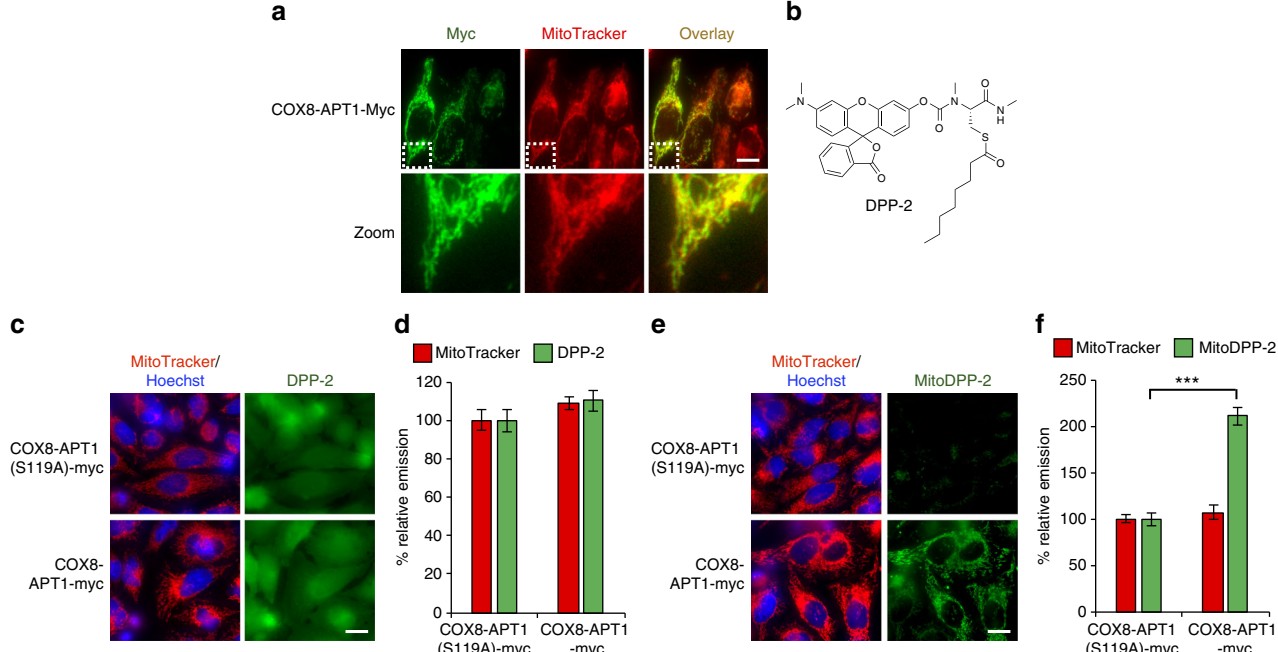

**Fig. 4** Overexpression of a mitochondrial-targeted APT selectively effects mitoDPP-2. **a** Immunostaining of HeLa cells overexpressing APT1 fused to a COX8 mitochondrial targeting sequence (COX8-APT1-myc) shows the fusion protein localized to mitochondria. **b** Structure of DPP-2, a previously reported, non-targeted cysteine deacylase probe. **c** HeLa cells transfected with COX8-APT1-myc or a catalytically inactive version, COX8-APT1(S119A)-myc, for 30 h, then loaded with 1 μM DPP-2 for 10 min, and analyzed by fluorescence microscopy. **d** Quantification of experiment described in (**c**) reveals that overexpressing a mitochondrial-targeted APT has no effect on the cytosolic APT signal as measured by DPP-2. **e** HeLa cells transfected with COX8-APT1-myc or a catalytically inactive version, COX8-APT1(S119A)-myc, for 30 h, then loaded with 500 nM mitoDPP-2 for 10 min, and analyzed by fluorescence microscopy. **f** Quantification of experiment described in (**e**) reveals that overexpressing a mitochondrial-targeted APT causes a significant enhancement on the mitochondrial APT signal as measured by mitoDPP-2. For all imaging, 20 μm scale bar shown. For plots, statistical analyses performed with a two-tailed Student's t-test with unequal variance, ***P value < 5 × 10⁻⁷; n = 8 images from two biological replicates, error bars are ± s.e.m

with MitoTracker (Fig. 3d, e, g, h, Supplementary Figs. 4 and 5). Altogether, these results indicate that mitoDPP-2 successfully localizes to the mitochondria and turns on in diverse cell lines.

We next tested whether the measured mitochondrial deacylase activity was enzymatic. As previously stated, there are no confirmed mitochondrial protein cysteine deacylases. Therefore, one possibility is that the cleavage of the thioester on the probe is non-enzymatic and mediated by the more basic environment and high thiol content within the mitochondria. To rule out this possibility, we deployed Palmostatin B (PalmB), a beta-lactone molecule that inhibits all known proteins with S-depalmitoylase activity, including APT1 and APT2[45], the ABHD17 proteins[12] and a host of other cellular "lipases". We reasoned that PalmB would also inhibit the mitochondrial cysteine deacylases, if there were any. Indeed, we found that pretreating the HEK293T, MCF-7, or A549 cells with 10 μM PalmB results in a blockage of at least 80% of the MitoDPP-2 signal (Fig. 3c, f, i and Supplementary Figs. 3, 4, 5). As a control, the level of fluorescence from MitoTracker was comparatively unaffected by PalmB treatment.

To confirm the measured cysteine deacylase activity reflects mitochondrial activity, we performed mitoDPP-2 imaging in isolated live, respiring mitochondria. The live mitochondria also showed PalmB-sensitive S-deacylation activity, confirming the presence of endogenous APTs in mitochondria (Supplementary Fig. 6). Finally, to rule out the possibility that the probe is reacting elsewhere in the cell prior to mitochondrial transport in live cells, we overexpressed APT1 with a mitochondrial localization sequence to see if overexpression of a mitochondrial APT selectively affected the mitoDPP-2 signal. As seen in Fig. 4a (Supplementary Figs. 7 and 8), fusion of a COX8 peptide

sequence[46] onto the N-terminus of APT1 resulted in mitochondrial localization of the protein. Overexpression of the COX8-APT1 fusion had no effect on the cytosolic APT signal, as measured by DPP-2, a non-targeted probe (Fig. 4b, c, d, Supplementary Fig. 9), but caused a substantial increase in the mitochondrial APT signal, as measured by mitoDPP-2 (Fig. 4e, f, Supplementary Fig. 10). A catalytically inactive APT1 fusion served as a control. Collectively, these experiments confirm that mitoDPP-2 measures mitochondrial APTs and reveal a previously unaccounted role for APTs in mitochondria. Therefore, we next sought to explore whether any of the known S-depalmitoylases have additional functions within the mitochondria.

**Screen reveals APT1 as putative mitochondrial S-deacylase**. In order to determine whether any of the known or putative S-depalmitoylases have additional roles in the mitochondria, we deployed mitoDPP-2 in a small RNAi screen. For the screen, we targeted the known cytosolic S-depalmitoylases APT1 and APT2, the lysosomal S-depalmitoylase PPT1[47], and the putative cytosolic S-depalmitoylases ABHD17A/B/C[12,13]. In addition, we included PPT2, a lysosomal thioesterase protein that has been shown to not have S-deacylation activity on peptide-based substrates[48], and LYPLAL1, a mitochondrial protein that is the most closely related member of the metabolic serine hydrolases to APT1 and APT2 with unknown function[49]. We cloned at least two shRNA vectors against each target, which were pooled together for the assay (Supplementary Tables 2 and 3). Using a 96-well plate reader-based assay, we found that only knockdown of APT1 decreased the signal from mitoDPP-2 (Supplementary Fig. 11), suggesting APT1 might be active in mitochondria.

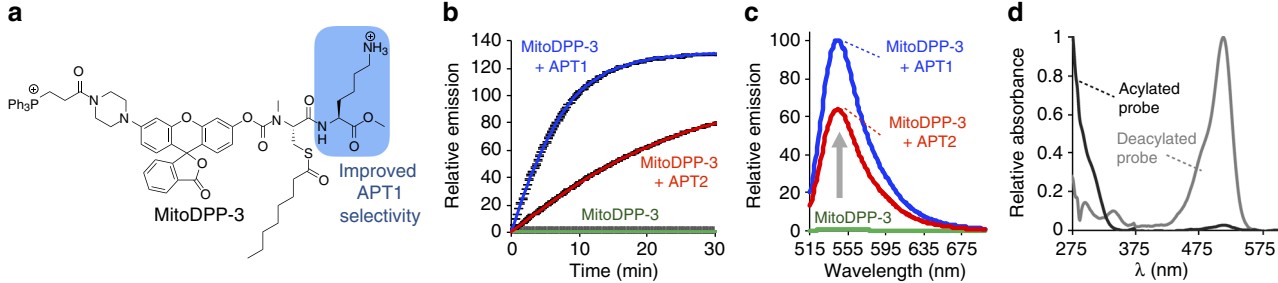

**Fig. 5** In vitro activity of mitoDPP-3. **a** Structure of mitoDPP-3. **b** In vitro fluorescence assay of 1 µM mitoDPP-3 in HEPES (20 mM, pH 7.4, 150 mM NaCl, 0.1% Triton X-100) with either 50 nM purified APT1 or APT2 ($\lambda_{ex}$ 490/20 nm, $\lambda_{em}$ 545/20 nm). For plots, $n = 3$, error bars are ± s.e.m. **c** Fluorescence emission spectra at 30 min from probes as treated in (**b**). **d** UV–vis spectra of 25 µM mitoDPP-3 (black; normalized at 275 nm) and deprotected fluorophore product (gray; normalized at 513 nm) in HEPES (20 mM, pH 7.4, 150 mM NaCl, 0.1% Triton X-100). MitoDPP-3 shows UV–vis absorbance at 300 nm with extinction coefficient $12.6 \times 10^3\,M^{-1}\,cm^{-1}$

**APT1 is an active mitochondrial S-deacylase**. MitoDPP-2 is a pan-activity S-deacylase probe, which reacts roughly equally well with APT1 and APT2 in vitro. Based on the genetic screen and follow-up APT1 inhibition studies (Supplementary Figs. 11a, b and 12a), we found that APT1 accounts for some, but not all, of the mitochondrial S-deacylase activity. Therefore, to boost the sensitivity for APT1, we designed mitoDPP-3 (Fig. 5a), which features a C-terminal lysine residue that we previously found increases the preference for APT1 (Supplementary Fig. 12c, d) over APT2[34]. Synthesis of mitoDPP-3 proceeded smoothly over five steps, and full synthetic procedures and chemical characterization are outlined in Supplementary Methods. MitoDPP-3 performed similarly to mitoDPP-2 in in vitro biochemical assays, with 1 µM of mitoDPP-3 showing a dramatic increase in fluorescence with 50 nM of recombinant human APT1 or APT2 (Fig. 5b–d). However, as designed, mitoDPP-3 slightly prefers APT1 over APT2 (Fig. 5b), very similar to the differences we observed with the cytoplasmic DPP-2 and DPP-3 probes developed previously.

MitoDPP-3, like mitoDPP-2, localizes to mitochondria and is sensitive to PalmB treatment (Fig. 6a–c, Supplementary Fig. 13). We deployed ML348 and ML349, selective inhibitors of APT1 and APT2[50–52], respectively, to test the effects of pharmacological inhibition of each APT on the mitoDPP-3 signal. APT1 inhibition blocked ~40% of the signal from mitoDPP-3 (Fig. 6d, Supplementary Figs. 12b and 14), while treatment with identical amounts of the APT2 inhibitor had no effect (Fig. 6e, Supplementary Fig. 15). Genetic perturbation yielded similar results, as we found that APT1 RNAi perturbed the mitoDPP-3 signal (Fig. 6f, Supplementary Table 3, Supplementary Figs. 16a and 17) while RNAi targeting APT2 had no effect (Fig. 6g, Supplementary Table 3 and Supplementary Fig. 18). Altogether, these data show that APT1 is active in mitochondria.

**APT1 is predominantly localized in mitochondria**. APT1 was annotated as a cytosolic protein when it was first reported twenty years ago[35]. Consistent with the original annotation, recent reports showed that overexpressed APT1 and APT2, tagged with mCitrine, associated with the Golgi apparatus and the plasma membrane[11,36]. Since we observed APT1's activity in mitochondria, we investigated if APT1 could localize there. We overexpressed a C-terminal myc-tagged APT1 and were surprised to find that it in fact localizes primarily to mitochondria (Fig. 7a). Given this observation was strikingly different from what was reported previously, we sought to uncover the discrepancy. We co-expressed our new APT1-myc fusion with the previously reported APT1-mCitrine fusion. Again, APT1-myc localized to mitochondria, while APT1-mCitrine was associated with the

Golgi apparatus and at the plasma membrane, as had been reported previously (Supplementary Fig. 19a). In contrast, we found APT2 associated with the Golgi apparatus and at the plasma membrane irrespective of the tag used (Supplementary Fig. 19b). We postulated that the fluorescent protein tag was perturbing APT1 trafficking, given the large size of the fluorescent protein tag relative to APT1. To test this, we inserted a stop codon at the end of APT1 in the APT1-mCitrine vector, before the start of the C-terminal mCitrine sequence. As postulated, this overexpressed, untagged APT1 also primarily localized to mitochondria, with some small fraction present in the cytosol (Supplementary Fig. 20).

Intrigued by the observation that APT1 expressed without a tag is primarily in the mitochondria, we next examined the localization of endogenous APT1. Standard immunostaining protocols could not detect endogenous APT1. Therefore, we used a more denaturing protocol, including incubation with SDS and heating the sample, inspired by antigen-retrieval techniques used for tissue immunohistochemistry. With this additional processing, we found endogenous APT1 localizes to mitochondria (Fig. 7b). To confirm the selectivity of the signal, we silenced APT1, which abrogated the APT1 staining. In addition, we performed similar staining experiments in HAP1 cells, both WT and CRISPR-Cas9 knockout for APT1 and APT2 (Supplementary Fig. 21). Again, we found that the APT1 signal was primarily associated with mitochondria (Supplementary Fig. 22). Finally, to further confirm the immunofluorescence results, we performed subcellular fractionation of HeLa cells (Supplementary Fig. 23), which showed that endogenous APT1 is highly enriched in purified mitochondria fractions, which are also highly enriched for TOM20, but devoid of alpha-tubulin (cytosol marker) and GM130 (Golgi marker) (Fig. 7c, Supplementary Fig. 24). Taken together, these experiments reveal that APT1 is in fact primarily localized at mitochondria, explaining the observed APT1 mitochondrial activity as measured by the mitoDPPs. Moreover, given that APT1 is naturally localized predominantly in mitochondria, the control experiments shown in Fig. 4 did actually not need additional mitochondrial localization tags.

**Palmitate increases mitochondrial S-deacylase activity**. Recently we discovered that cytosolic S-deacylases respond to growth signaling by transient inhibition of their S-deacylase activity[34]. We sought to test whether the mitochondrial S-deacylases are constitutively active, or whether we could determine biological conditions that selectively affect mitochondrial S-deacylase activity levels. If the mitochondrial APTs respond dynamically to local lipid levels, this would suggest mitochondrial S-deacylases are regulated.

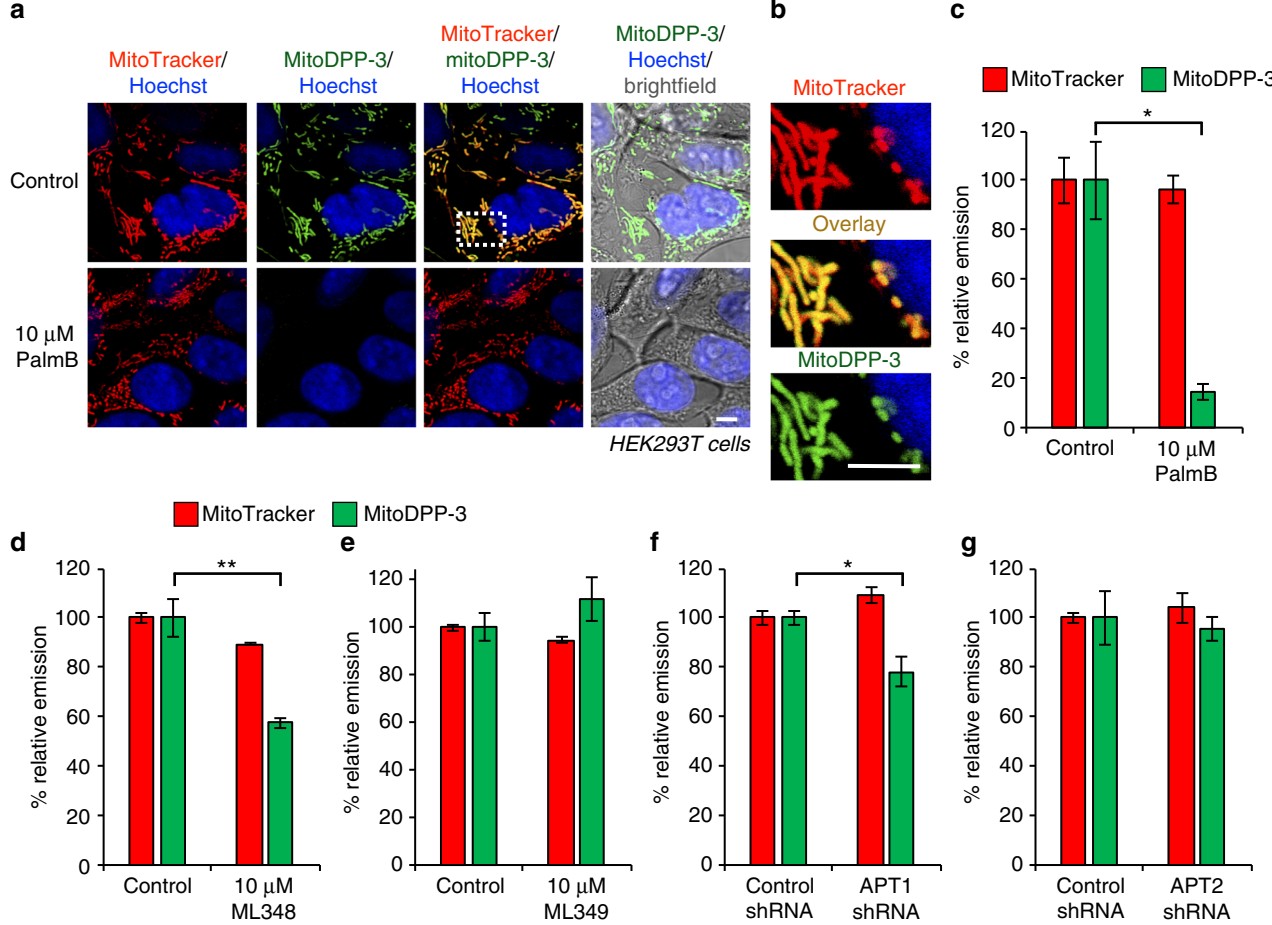

**Fig. 6** MitoDPP-3 reveals APT1 activity in mitochondria. **a** HEK293T cells treated for 30 min with 1 μM Hoechst 33342, 100 nM MitoTracker Deep Red, and either DMSO or 10 μM PalmB for 30 min, washed, loaded with 324 nM mitoDPP-3 for 10 min, and then analyzed by confocal fluorescence microscopy. 5 μm scale bars shown. **b** Enlarged portion of image indicated by dotted white box in (**a**) showing colocalization of mitoDPP-3 with MitoTracker. Quantification of the relative fluorescence intensity from mitoDPP-3 and MitoTracker in either control or PalmB-treated cells (**c**). ML348-treated cells (**d**), ML349-treated cells (**e**), cells with APT1 knocked down (**f**), or cells with APT2 knocked down (**g**). For all plots, statistical analyses performed with a two-tailed Student's $t$-test with unequal variance, *$P$ value < 0.05; **$P$ value < 0.009, $n = 3$ for (**c**), $n \geq 4$ for (**d**–**g**), error bars are ± s.e.m

We decided to first test whether treatment with palmitate, which causes cell stress in part through mitochondrial metabolic signaling[53], changed the levels of mitochondrial cysteine deacylase activity. Indeed, starvation of HEK293T cells for 6 h followed by transient treatment with 1% BSA ± 1 mM palmitate for 6 h shows a significant increase in mitochondrial cysteine deacylase activity in the palmitate-treated samples, as measured by mitoDPP-2 (Fig. 8, Supplementary Information Fig. 25). Aside from changes to mitochondrial metabolism, palmitate stimulation may also change mitochondrial palmitoyl-CoA levels, the source of protein S-palmitoylation, resulting in an elevation in the rate of proteome lipidation, which is counterbalanced by activating S-deacylases in the organelle. Further experiments are needed to deduce these downstream effects, but for our purposes, the observation that lipid stress causes a change in mitochondrial S-deacylase activity suggests that it may be possible to find genetic perturbations that selectively effect the mitochondria through local lipid regulation.

**APTs respond locally to ACOT1 and ACOT11 knockdown.** Given the response to bolus palmitate treatment, we sought to test the subcellular regulation of APTs by perturbation of organelle-specific lipid pools. We decided to focus our attention on acyl-

CoA thioesterases (ACOTs), which regulate cellular pools of acyl-CoA donors through thioesterase activity, generating heat, lipids, and CoA in the process[54,55]. Different ACOTs selectively effect different pools of acyl-CoA donors throughout the cell. We therefore reasoned we might be able to find ACOTs that selectively influence either the cytoplasmic or mitochondrial S-deacylase levels.

We generated a library of pooled shRNAs targeting 10 human ACOT family members to individually knockdown each protein (Supplementary Table 2). Using our plate reader screen, we measured the effects of ACOT knockdown on both cytosolic and mitochondrial S-deacylase activity levels using our previously developed cytosolic probe DPP-2 and our new mitochondrial probe mitoDPP-2, respectively (Supplementary Fig. 26). We found that while knockdown of ACOT1 elevated the signal from DPP-2, knockdown of ACOT11 increased the signal from mitoDPP-2. ACOT1, a cytosolic protein, prefers long saturated and monounsaturated acyl-CoA substrates[55]. In addition, ACOT1 has been shown to regulate the ligand availability for nuclear hormone receptor PPAR(alpha) and hepatocyte nuclear factor (HNF4) alpha through modulation of the cytoplasmic pool of long-chain acyl-CoA donors[56]. ACOT11 (also known as BFIT and Them1) knockout in mice was recently shown to promote resistance to diet-induced obesity despite greater food

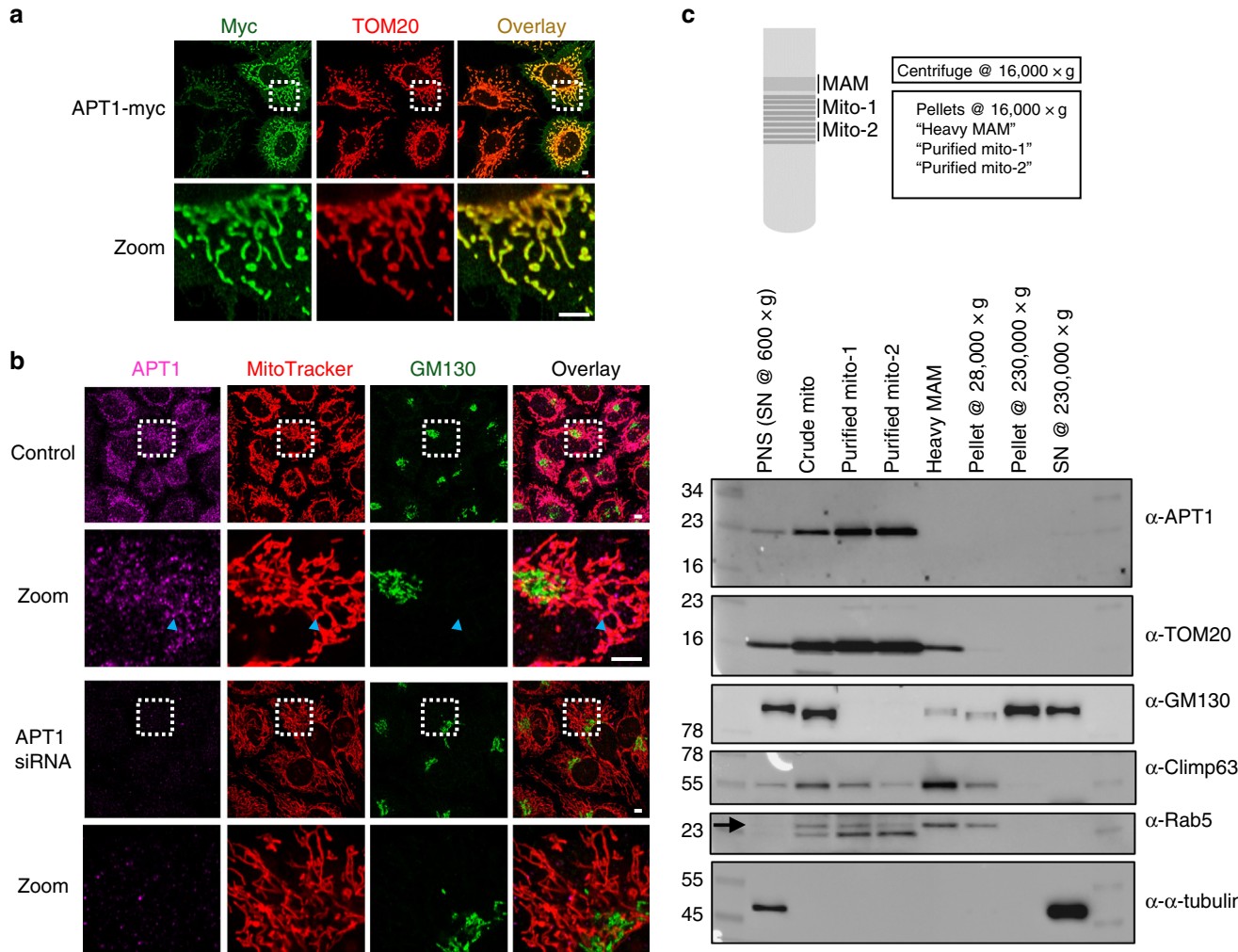

**Fig. 7** Immunostaining and subcellular fractionation experiments reveal APT1 is localized in mitochondria. **a** Immunostaining of HeLa cells overexpressing APT1 with a myc tag reveals APT1 localized to mitochondria. 5 µm scale bars shown. **b** Immunostaining of endogenous APT1 in HeLa cells confirms APT1 is predominantly localized in mitochondria mitochondria (e.g., co-localization pointed by the blue arrows). Knockdown of APT1 by RNAi confirms that the measured signal is due to APT1. 5 µm scale bars shown. **c** HeLa cells were fractionated to isolate mitochondria by means of differential centrifugation and a continuous Percoll gradient[63] (Supplementary Fig. 23). Fractions recovered at different steps of the protocol were loaded in an SDS-polyacrylamide gel (10 µg of total protein per lane) and analyzed by western blot. The fractions that are highly enriched for mitochondria (high Tom20, low alpha-tubulin and GM130) display the highest levels of APT1, confirming the immunostaining experiments and demonstrating APT1 is predominantly localized in mitochondria. In the anti-Rab5 panel, the upper bands (arrow) correspond to Rab5. The lower bands in this panel correspond to Tom20 staining since the nitrocellulose membrane used for Tom20 staining was reblotted with anti-rab5 antibody. Similar enrichment patterns were observed for all antibodies in at least four independent fractionation experiments. Numbers on the left indicate protein sizes of the corresponding bands of the molecular weight marker (first lane). PNS post-nuclear supernatant, MAM mitochondria-associated ER membranes, SN supernatant. Original uncropped blots are provided in Supplementary Fig. 24

consumption[57]. This effect was partially attributed to perturbations in mitochondrial lipid homeostasis due to loss of ACOT11. Due to the differentially localized *S*-deacylase response to ACOT1 and ACOT11 perturbation and the literature precedent that supports our model, we decided to validate these observations further.

To confirm the effects measured by the crude high-throughput plate reader assay, we performed a series of cell imaging experiments with both DPP-2 and mitoDPP-2. The live cell imaging confirmed the genetic screening results, revealing that ACOT11 knockdown has no effect on DPP-2 (Fig. 9a, b; Supplementary Fig. 27), but dramatically increases the signal from mitoDPP-2 without affecting the signal from MitoTracker (Fig. 9c, d; Supplementary Fig. 28). In comparison, ACOT1

knockdown increases the signal from DPP-2 (Fig. 9e, f; Supplementary Fig. 29), but not mitoDPP-2 (Fig. 9g, h; Supplementary Fig. 30). Altogether, these data reveal that cells tune their mitochondrial deacylase activity in response to the metabolic state of the cell. Moreover, the differential response of ACOT1/11 perturbation on DPP-2/mitoDPP-2 further confirms the preferential localization of mitoDPPs in mitochondria (Fig. 10).

## Discussion

The expansion of the functional roles of APT1 to include regulation of the *S*-palmitoylation status of the mitochondrial proteome opens up new biological ramifications for this key "eraser"

protein. Two recent quantitative proteomic studies that revealed mitochondrial localization of APT1 analogs in two organisms, *Trypanosoma brucei*[58] and *S. cerevisiae*[59], complement our

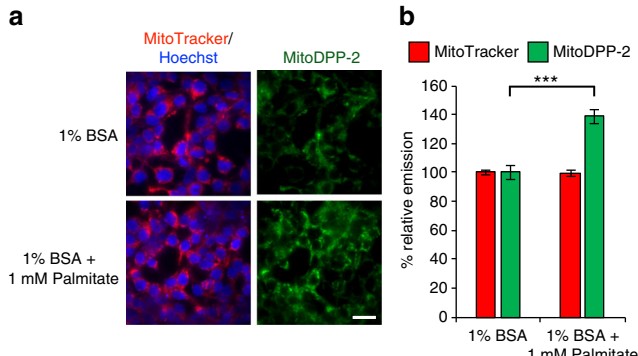

**Fig. 8** Palmitate activates mitochondrial *S*-deacylase activity. **a** HEK293T cells were treated for 6 h with 1% BSA ± 1 mM palmitate post-starvation (6 h), followed by treatment with 1 μM Hoechst 33342, 100 nM MitoTracker Deep Red for 30 min, washed, loaded with 500 nM mitoDPP-2 for 10 min, and then analyzed by fluorescence microscopy. 20 μm scale bar shown. **b** Quantification of the relative fluorescence intensity from mitoDPP-2 and MitoTracker in 1% BSA ± 1 mM palmitate-treated cells from (**a**). Statistical analyses performed with a two-tailed Student's *t*-test with unequal variance, **P value < 0.0005, n = 6 for (**b**), error bars are ± s.e.m

findings here. Identification of the mitochondrial proteomic targets of APT1 is the next step toward understanding the functional consequences of APT1 mitochondrial activity. However, even though we discovered APT1 is predominantly localized in mitochondria, either pharmacological inhibition or genetic perturbation of APT1 will perturb the function of APT1 throughout the cell, convoluting the mitochondrial-specific regulatory effects. Therefore, organelle-targeted inhibitors are needed to separate the cytosolic and mitochondrial roles for this and other *S*-acylation erasers, which we are currently developing.

Previously, we observed that DPP-2, the simplest probe substrate with a methylamide C-terminal modification, reacted with APT1 robustly in vitro but was relatively insensitive to APT1 perturbation in live cells. Here we show that mitoDPP-2, which is based on DPP-2, shows similar activities as DPP-2 with APT1 and APT2 in vitro, but is sensitive to APT1 perturbation in live cells. The opposite cellular selectivity of DPP-2 and mitoDPP-2 suggests that cellular localization is a key determinant of APT target engagement. As we show here, the overall *S*-deacylation activity within a particular region of the cell can be altered depending on cell state. Therefore, it seems reasonable that different *S*-deacylases could be implicated in regulating the same targets depending on the specific cell state, tissues type, or disease state.

That mitoDPP-2 and mitoDPP-3 signals are inhibited substantially more by PalmB treatment than either APT1 inhibition with a small molecule or by RNAi suggests there may be other mitochondrial *S*-deacylases. We tested all known and putative *S*-

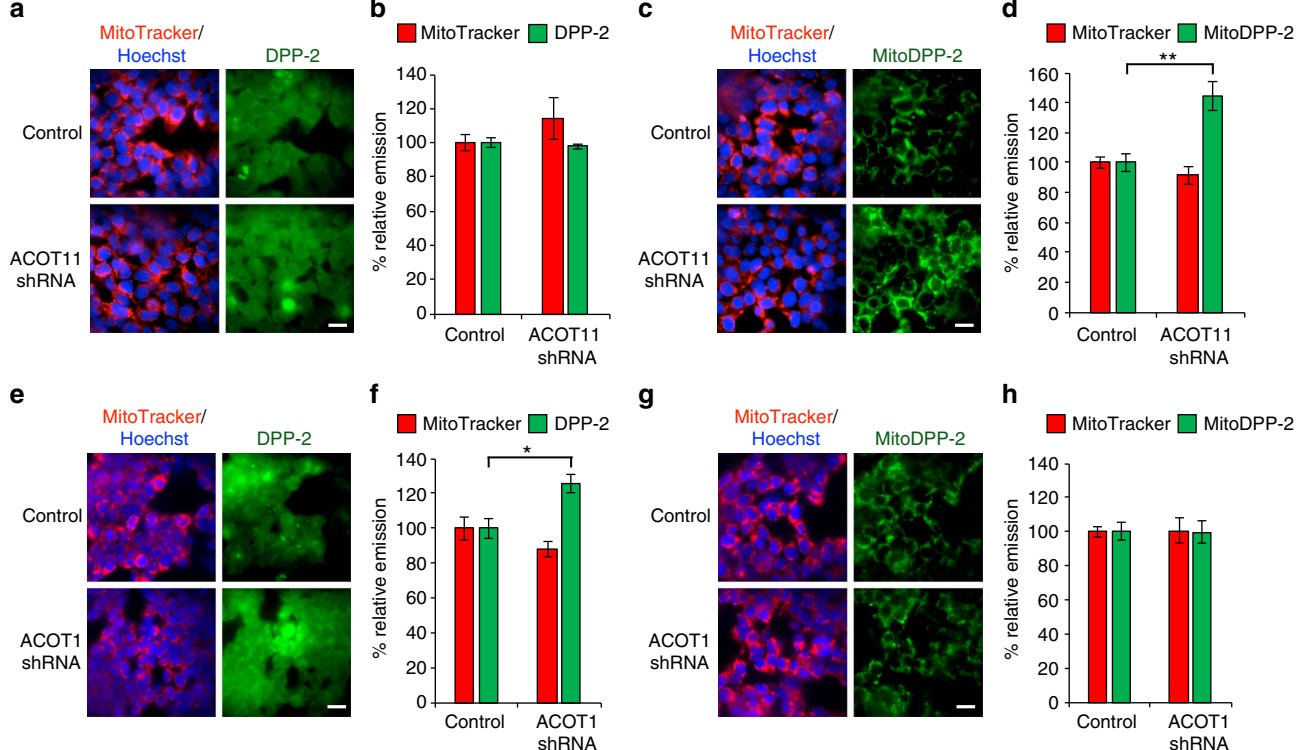

**Fig. 9** ACOT11 and ACOT1 knockdown selectively perturbs mitochondrial and cytosolic *S*-deacylase activities. After transfection with either control or ACOT11 or ACOT1 shRNA vectors, HEK293T treated with 1 μM Hoechst 33342, 100 nM MitoTracker Deep Red for 30 min, washed, loaded with 1 μM DPP-2 (**a**, **e**) or 500 nM mitoDPP-2 (**c**, **g**) for 10 min, and then analyzed by fluorescence microscopy. **b** Quantification of experiment shown in (**a**). ACOT11 perturbation does not affect DPP-2 (cytosolic) signal. **d** Quantification of experiments shown in (**c**). ACOT11 knockdown causes an increase in mitoDPP-2 (mitochondrial) signal. **f** Quantification of experiment shown in (**e**). ACOT1 knockdown causes an increase in DPP-2 (cytosolic) signal. **h** Quantification of experiments shown in (**g**). ACOT1 perturbation does not affect mitoDPP-2 (mitochondrial) signal. For all imaging, 20 μm scale bar shown. For all plots, statistical analyses performed with a two-tailed Student's *t*-test with unequal variance, *P value < 0.05; **P value < 0.007, n = 5 for (**a**, **c**, **e** and **g**), error bars are ± s.e.m

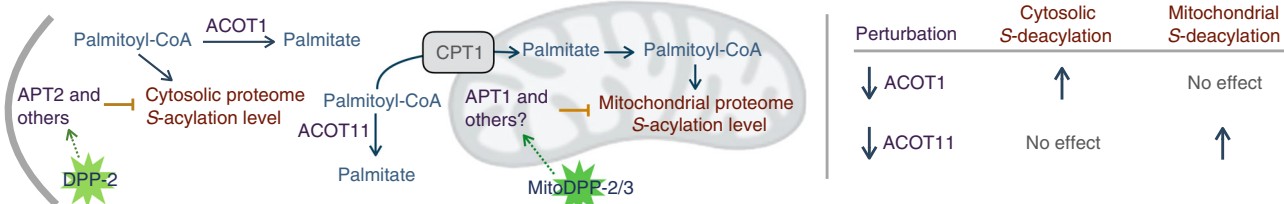

**Fig. 10** Schematic of results from this study. Of the identified APTs, APT1 localizes predominantly to mitochondria, whereas APT2 is found in the cytosol. Cytosolic vs. mitochondrial APT activity can be differentially increased by knockdown of ACOT1 or ACOT11, respectively

depalmitoylases in our RNAi screen, and found that only APT1 contribute significantly to the mitoDPP-2 signal. Moreover, even the signal from mitoDPP-3, which is an improved probe for APT1, is not inhibited by APT1 shRNA or APT1 inhibitors to the same degree as PalmB, further suggesting that there might be other mitochondrial S-deacylases yet to be discovered.

One caveat is that there is a possibility that enzymes that are not APTs can process the mitoDPPs presented here, giving false signals. There is also a possibility that the current scaffolds may not be substrates for some APTs, leading to false negative signals as well. Recently, we were able to improve the DPP design and accommodate natural lipids[38], which could lead to mitoDPPs with improves APT selectivity. In addition, we recently developed a ratiometric, "RDP" scaffold using a different sensing mechanism, which permits more quantitative APT activity measurements[37]. None-the-less, the genetic screening approaches developed and validated here provide a path forward toward mining the proteome for additional enzymes that either regulate or mediate mitochondrial S-deacylation.

ACOT11 was recently found to interact with mitochondria to control local fluxes of long-chain fatty acids in that compartment[60]. The discovery that perturbation to this regulator of mitochondrial lipid levels changes the mitochondrial APT levels suggests that mitochondrial deacylation has consequences to cell signaling. Discovering the other players in mitochondrial S-acylation signaling, such as the acyl-transferases(s)[61], will aid in testing the physiological relevance of this observed signal. The observation that cytosolic lipid perturbation through ACOT1 knockdown had no effect on the mitochondria but did change cytosolic APT levels is also interesting. The mutually-exclusive effects illustrate how tightly lipid pools, and the enzymes that regulate them, are controlled in different compartments. It is not obvious why the APTs respond to these particular lipid alterations with enhanced activity, but it suggests the dynamic equilibrium between installation and removal of the lipid modification is important and regulated.

More broadly, the development of spatially-constrained biochemical activity probes with genetic screens and pharmacological perturbation provides a powerful approach to study biological signaling that is mediated by subcellular distribution. The fact that we were able to uncover the correct APT1 localization using chemical tools validates the complementarity between genetic and cell biology approaches with chemical biology approaches. Spatial regulation is exceptionally challenging to probe genetically, as knockdown affects the protein everywhere in the cell. Our mitoDPP platform not only provides critically needed tools to study mitochondrial lipid signaling, but can also be immediately expanded to probe S-deacylation in other cellular compartments and to probe other mitochondrial cysteine eraser enzymes.

## Methods
**General materials and methods**. DMEM Glutamax (10569-010, Gibco), FBS (Gibco/Life Technologies, Qualified US origin), Live Cell Imaging Solution

(A14291DJ, Molecular Probes), Opti-MEM (31985-070, Gibco), Lipofectamine 3000 reagent (100022052, Invitrogen) MitoTracker Deep Red FM (M22426, Invitrogen), Hoechst 33342 (Fisher), were purchased as mentioned in parenthesis. Silica gel P60 (SiliCycle, 40–63 μm, 230–400 mesh) was used for column chromatography. Analytical thin layer chromatography was performed using SiliCycle 60 F254 silica gel (precoated sheets, 0.25 mm thick). All chemicals for synthesis were purchased from Sigma-Aldrich (St. Louis, MO) or Fisher Scientific (Pittsburgh, PA) unless otherwise noted and used as received. ML348 and ML349 were purchased from Tocris (Bristol, UK). Full description of syntheses of mitoDPP-2 and mitoDPP-3 are provided in Supplementary Information and Supplementary Figs. 31 and 32, and characterization of all final products and intermediates shown in Supplementary Figs. 33–45 $^{1}$H NMR and $^{13}$C NMR spectra were collected in NMR solvents CDCl$_3$ (Sigma-Aldrich, St. Louis, MO) at 25 °C using a 500 MHz Bruker Avance II+spectrometer with 5 mm QNP probe or 400 MHz bruker DRX400 at the Department of Chemistry NMR Facility at the University of Chicago. $^{1}$H-NMR chemical shifts are reported in parts per million (ppm) relative to the peak of residual proton signals from (CDCl$_3$ 7.26 ppm or CH$_2$Cl$_2$ peak at 5.30 ppm). Multiplicities are given as: s (singlet), d (doublet), t (triplet), q (quartet), dd (doublet of doublets), m (multiplet). $^{13}$C-NMR chemical shifts are reported in parts per million (ppm) relative to the peak of residual proton signals from (CDCl$_3$ 77.16 ppm). Analysis of NMR was done in iNMR version 5.5.1. and NMR plots were obtained from Topspin 2.1. High resolution mass spectra were obtained on a Agilent 6224 TOF High Resolution Accurate Mass Spectrometer (HRA-MS) using combination of APCI and ESI at the Department of Chemistry Mass Spectrometry Facility at the University of Chicago. Low resolution mass spectral analyses and liquid chromatography analysis were carried out on an Advion Expression-L mass spectrometer (Ithaca, NY) coupled with an Agilent 1220 Infinity LC System (Santa Clara, CA). In vitro biochemical assays with purified APT1/APT2 enzymes and high-throughput screening of shRNA libraries were performed on Biotek synergy Neo2 plate reader. The concentration of the DMSO stock of mitoDPP-3 was normalized with a known concentration of mitoDPP-2 by absorbance at 300 nm in ethanol.

**shRNA library cloning**. shRNA plasmids were constructed in pLKO.1 vector[62] by restriction cloning adapting the protocol available on Addgene. Two shRNAs were constructed for each target (except, three for APT1) using the Broad Institute's Online Genetic Perturbation Platform database. The sequences for each target are shown in Supplementary Table 2. Sequencing was validated at the University of Chicago Comprehensive Cancer Center DNA Sequencing and Genotyping Facility. For experiments, the shRNA clones (50 ng/μL) were pooled. The pLKO.1 vector was used as a control shRNA against shRNAs for the targets of interest.

**In vitro kinetic assays of mitoDPP-2 and mitoDPP-3**. In total 100 μL of 3 μM mitoDPP-2 or mitoDPP-3 in HEPES buffer (20 mM, pH = 7.4, 150 mM NaCl, 0.1% Triton X-100) were added to a 96-well optical bottom plate (Nunc 265301, Thermo Scientific) at room temperature. 200 μL of either HEPES buffer alone or HEPES buffer containing 75 nM APT1 or APT2 (purification using previous methods[34]) was added using a multi-channel pipette, resulting in a final concentration of 1 μM mitoDPP-2/mitoDPP-3 and 50 nM APT1 or APT2. Fluorescence intensities ($\lambda_{ex}$ 490/20 nm, $\lambda_{em}$ 545/20 nm, Gain 50, read from bottom with height 4.5 mm, and sweep method) were measured at 15 s time intervals for 30 min at 37 °C. Following the 30 min kinetic run, emission spectra were obtained ($\lambda_{ex}$ 480/20 nm, $\lambda_{em}$ 515–700 nm, Gain 60, read from bottom with height 4.5 mm, and sweep method).

**Enzyme kinetics of mitoDPP-2 and mitoDPP-3 with APT1**. In total 150 μL of 1 M DTT in HEPES buffer (20 mM, pH = 7.4, 150 mM NaCl, 0.1% Triton X-100) was added to 150 μL of different concentrations of mitoDPP-2 or mito-DPP-3 and then the stable fluorescence intensities of totally deacylated product were measured by the methods described above. Fluorescence intensity of per molar deacylated-product was the average under different concentrations. For Michaelis–Menten constant, in vitro kinetics curves were obtained for different final concentrations of mitoDPP-2 and mitoDPP-3 in HEPES buffer with 50 nM APT1. The fluorescence intensities measured were then normalized to the concentrations of product by fluorescence-intensity-per-molar-product. A Lineweaver–Burk plot was used to

calculate the Michaelis constant, $K_M$, and the maximum velocity, $V_{max}$, which is proportional to $k_{cat}$ by concentration of APT1. All fits have $R^2$ values of > 0.99.

**In vitro kinetic of mitoDPP-2 at mitochondrial pH 8.0.** In total 150 µL of 2 µM mitoDPP-2 in HEPES buffer (20 mM, 150 mM NaCl, pH = 8.0) were added to a 96-well optical bottom plate (Nunc 265301, Thermo Scientific) at room temperature. A total of 150 µL of either HEPES buffer alone or HEPES buffer containing 100 nM APT1, was added using a multi-channel pipette, resulting in a final concentration of 1 µM mitoDPP-2 and 50 nM APT1. Fluorescence intensities were measured by the methods described above.

**UV-visible spectroscopy.** UV-Vis measurements were carried out on a Shimadzu UV-2700 UV–vis spectrophotometer. UV–vis absorption measurements of 25 µM of mitoDPP-2, mitoDPP-3 and the deacylated fluorophore product (i.e., phosphonium rhodol) were made in HEPES (20 mM, pH = 7.4, 150 mM NaCl) containing 0.1% Triton X-100 and placed in micro black 8 cell chamber with 10 mm optical path length (Shimadzu). For extinction coefficient measurements HEPES (20 mM, pH = 7.4, 150 mM NaCl) containing 0.1% Triton X-100 was used to make dilute solution of mitoDPP-2, mitoDPP-3 and the deacylated fluorophore product. Extinction coefficients were calculated at maximum $\lambda_{abs}$ by linear least-squares fitting of plots of A vs. concentration. All fits gave $R^2$ values of ≥ 0.999.

**Cell culture and maintenance.** HEK293T (ATCC), MCF-7 cells (from Cellular Screening Center, University of Chicago), A549 cells (gift from Prof. Yamuna Krishnan), and HeLa cells (ATCC) were maintained in DMEM Glutamax (10% FBS, 1% P/S) at 37 °C and 5% $CO_2$. HEK293T and MCF-7 cells are listed in the database of commonly misidentified cell lines maintained by ICLAC (http://iclac.org/databases/cross-contaminations/). We obtained fresh cells from ATCC or early passage aliquots from the Cellular Screening Center, University of Chicago, which were frozen down at an early passage (passage 6) in individual aliquots. The cells were then used for less than 25 passages for all experiments. Multiple biological replicates were performed with cells from different passages and freshly thawed aliquots. There was no testing for mycoplasma infection or further authentication because early passage cells were used for all experiments.

**Confocal Imaging of mitoDPP-2 and mitoDPP-3.** HEK293T cells ($7.5$–$8.0 \times 10^4$/ well), MCF-7 cells (75000/well) and A549 cells (35000/well) were plated in 500 µL DMEM glutamax (10% FBS) into two wells of a four well chambered imaging dish (D35C4-20-1.5-N, Cellvis), which were precoated with 4 µg Poly-D-lysine (30–70 KDa, Alfa Aesar) for 2 h before plating. After 24–27 h of cell culturing, the media was replaced with 1 µM Hoechst 33342, 100 nM MitoTracker Deep Red and DMSO/10 µM PalmB in 400 µL DMEM glutamax. After 30 min of incubation at 37 °C, the cells were washed with 400 µL of Live Cell Imaging Solution and replaced by 250 nM mitoDPP-2/3 in Live Cell Imaging Solution. After 10 min of incubation at 37 °C, images were obtained on a confocal microscope (Leica SP5 tandem scanner spectral with custom fit incubation jacket) with ×100 oil objective (HCX PL APO, N/A 1.46), pinhole (0.5 Airy unit), scan speed (8000 Hz) and line average (48) attached to detectors PMT (for Hoechst) and HyD (for mitoDPPs and Mito-Tracker). Lasers 405 nm, 514 nm, and 633 nm were used for Hoechst, mitoDPPs and MitoTracker, respectively. See Supporting information (Supplementary Table 4) for further details on laser intensity and gain values used for imaging. Analyses were performed in ImageJ (Wayne Rasband, NIH). For data analysis, the average fluorescence intensity per image in each experimental condition was obtained by gating signal on the MitoTracker signal, applying that in the corresponding mitoDPP-2/mitoDPP-3 image, and the data normalized to the average fluorescence intensity of the DMSO control.

**Mitochondrial targeted APT1.** The following sequence of Cox8-APT1 fusion was used to generate a mitochondrial-localized APT1. The N-terminal sequence of APT1 is underlined. MSVLTPLLLRGLTGSARRLPVPRAKIHSLGDPPVAT MCG…

**Immunofluorescence with mitochondrial-targeted APT1.** Approximately 40,000 HeLa cells/well were plated in 450 µL DMEM glutamax (10% FBS) into an 8 well dish (154534, Lab-Tek II Chamber slide). After 20–22 h, 125 µL of growth media was removed and cells were transfected with 175 ng of either plasmid following manufacture's conditions. Briefly, 7.5 µL of opti-MEM containing 0.53 µL of Lipofectamine 3000 was added to mix of 3.96 µL opti-MEM, 0.35 µL P3000 and 3.5 µL DNA (50 ng/µL), and the resulting DNA:Lipofectamine mix was incubated at room temperature for 15 min. After incubation, 15 µL of the DNA:Lipofectamine mix was added to the corresponding well of 8 well dish (154534, Lab-Tek II Chamber slide). For immunofluorescence analysis growth media was replaced 26 h post-transfection with 240 µL of DMEM glutamax (10% FBS) containing 300 nM MitoTracker Deep Red and 1 µM Hoechst 33342. After 30 min of incubation at 37 °C, the cells were washed with 240 µL of 1× PBS (3P399-500, Fischer) and 240 µL of 4% paraformaldehyde solution in 1× PBS was added. After incubation for 20 min at room temperature, the cells were washed with ice cold 5 × 400 µL PBS each time for 5 min. After washing, 240 µL of permeabilizing/blocking solution (PBS

containing 3% BSA and 0.1% saponin) was added and the cells were incubated at 4 °C for 1 h followed by incubation at room temperature for another 1 h. The permeabilizing/blocking solution was replaced with 240 µL of permeabilizing/blocking solution containing 1:2500 mouse monoclonal anti-c-myc (9E10, sc-40, SCBT). After overnight incubation at 4 °C, the cells were washed with ice cold 5 × 400 µL PBS each time for 5 min. The cells were then incubated with 1:4000 rabbit anti-mouse Alexa Fluor 488 IgG (H + L, A11059, Invitrogen) for 1 h at room temperature, followed by washing with ice cold 5 × 400 µL PBS, each time for 5 min. The cells were then sealed with cover glass #1.5 (48393-195, VWR) following addition of Prolong Diamond Antifade (P36961, Thermo Fischer). After 30 min, images were obtained on an inverted epifluorescence microscope (Lieca DMi8) equipped with a camera (Hamamatsu Orca-Flash 4.0) with ×63 oil objective (N/A 1.4) and light source (Sutter Lamda XL, 300 W Xenon) for GFP (ET 490/20x, Quad-S, ET 525/36 m), Hoechst 33342 (ET 402/15x, Quad-S, ET 455/50 m), MitoTracker (ET 645/30x, Quad-S, ET 705/72 m) and brightfield using Leica LASX software. Analyses were performed in ImageJ (Wayne Rasband, NIH).

**Fluorescence imaging with mitochondrial-targeted APT1.** In total 40,000 HeLa cells/well were plated in 450 µL DMEM glutamax (10% FBS) into an 8 well dish (C8-1.5H-N, CellVis). Transfection was carried out as described above. After 30 h of post-transfection the growth media was replaced with glutamax (10% FBS) containing 100 nM MitoTracker Deep Red and 1 µM Hoechst 33342. After 30 min of incubation at 37 °C, the cells were washed with 400 µL of Live Cell Imaging Solution and replaced by 1 µM DPP2/500 nM mitoDPP-2 in Live Cell Imaging Solution. After 10 min of incubation at 37 °C, images were obtained on an inverted epifluorescence microscope (Lieca DMi8) equipped with a camera (Hamamatsu Orca-Flash 4.0) with ×63 oil objective (N/A 1.4) and light source (Sutter Lamda XL, 300 W Xenon) for DPPs (YFP filter cube 1525306), Hoechst 33342 (ET 402/15x, Quad-S, ET 455/50 m), MitoTracker (ET 645/30x, Quad-S, ET 705/72 m) and brightfield using Leica LASX software. Analyses were performed in ImageJ (Wayne Rasband, NIH). For data analysis, the average fluorescence intensity per image in each experimental condition was obtained by gating cells on brightfield and applying that in the corresponding MitoTracker and DPP-2/mitoDPP-2 image. Eight images from two biological replicates were used to quantify the images, and the data was normalized to the average fluorescence intensity of the inactive mutant.

**Isolation and imaging live mitochondria.** Live mitochondria were isolated from fresh adult mouse liver that was minced and homogenized in a dounce homogenizer on ice in MSHE Buffer (210 mM mannitol, 70 mM sucrose, 5 mM HEPES, 1 mM EGTA, 0.5% FA-free BSA). The homogenate was centrifuged for 10 min at 4 °C at 800 g. Following careful aspiration of the fat layer, the supernatant was decanted through a filter and centrifuged twice more at 8000 g for 10 min. The final pellet containing mitochondria was resuspended in 600 µL of MSHE buffer and protein was quantified by BCA assay. Respiratory capacity of isolated mitochondria was assessed using the XF96 Seahorse Analyzer. 5 µL of mitochondria and 5 µL of MSHE buffer containing 200 nM MitoTracker Deep Red and 40 µM of PalmB were mixed and incubated at 37 °C for 20 min. 10 µL of 2 µM mitoDPP-2 in MSHE buffer was added and 10 µL of the mixture was transferred to the imaging dish (D35-20-1.5-N, CellVis) and covered with cover glass slip. After 10 min of incubation at room temperature, images were obtained on an inverted epifluorescence microscope (Olympus IX83) attached with EMCCD camera (Photometrics Evolve Delta) with ×100 oil objective (N/A 1.4) for mitoDPP-2 (excitation filter 500/20, dichroic chroma 89007, emission filter 535/30, exposure time 300 ms, EM gain 150), MitoTracker Deep Red (excitation filter 640/30, emission filter 705/72, dichroic chroma 89016, exposure time 100 ms, EM gain 75) and brightfield (exposure time 50 ms, EM gain 30). Analyses were performed in ImageJ (Wayne Rasband, NIH). For data analysis, the average fluorescence intensity per image in each experimental condition was obtained by selecting the whole image for both MitoTracker and mitoDPP-2, and the data were normalized to the average fluorescence intensity of the DMSO control. Two biological replicates with similar results were performed.

**DPP-2 and mitoDPP-2 plate reader assay with shRNA library.** In total 12,000–16,000 HEK293T cells/well were plated in 100 µL DMEM glutamax (10% FBS) into 96-well plate (P96-1-N, Cellvis), which was precoated with 3.75 µg Poly-D-lysine (30–70 KDa, Alfa Aesar) for 150 min. After 16–20 h, cells were transfected with 80 ng of either control shRNA for every 1st well of the row or shRNAs for the targets of interest following manufacture's conditions. Briefly, 4.4 µL of opti-MEM containing 0.4 µL of Lipofectamine 3000 was added to mix of 4 µL opti-MEM, 0.16 µL P3000 and 1.6 µL shRNAs mix (50 ng/µL), and resulting DNA:Lipofectamine mix was incubated at room temperature for 13–15 min. After incubation 9.7 µL of the DNA:Lipofectamine mix was added to the corresponding well of the 96-well dish. After 52–56 h from transfection, the cells were washed with 150 µL of Live Cell Imaging Buffer (ThermoFisher) and 70 µL of 1 µM DPP-2 or 500 nM mitoDPP-2 in Live Cell Imaging Buffer was added to the each of the eight wells per row using a multi-channel pipette. The cells were then incubated at room temperature for 10 min and total fluorescence intensity ($\lambda_{ex}$ 490/20 nm, $\lambda_{em}$ 545/20 nm, Gain 130, read from bottom with height 4.5 mm, and sweep method) was

measured at room temperature by scanning the area of each well with a matrix size 11 × 11. The data were normalized to the average fluorescence intensity of the control shRNA.

**Flow cytometry**. In total 1,000,000 HEK293T cells were plated on a 6 well dish (Denville Scientific Inc.) with 2 mL of DMEM glutamax (10% FBS). After 24–28 h, the cells were washed with DPBS and trypsinized with 150 μL 0.025% Trypsin. 1.5 mL of DMEM glutamax (10% FBS) was added and the cells were transferred to 2.0 mL tubes and then spun down at 135 rcf for 3 min. The media was discarded and the cells were resuspended in 1 mL of Live Cell Imaging Solution, and then spun down at 135 rcf for 3 min. The cell pellets obtained were resuspended in 1 mL of Live Cell Imaging Solution and divided into two 200 μL aliquots. To the aliquots, 1 mL of Live Cell Imaging Solution was added and the cells were treated with either DMSO carrier or 1 μM ML348. After 30 min of incubation at room temperature, the cells with each condition were divided into four 200 μL aliquots. To these cells, 50 μL of 5 μM DPP-2/10 μM DPP-3/2.5 μM mitoDPP-2/2.5 μM mitoDPP-3 in Live Cell Imaging Solution was added. After 10 min incubation at room temperature, the fluorescence signal in FITC channel was measured for 5000–10000 cells on a LSR-Fortessa 4–15 HTS (BD digital instrument, FSC 10 V, SSC 200 V, 488 nm laser with 530/30 nm filter for FITC 400 V). Data were analyzed by FlowJo software version 10.0.8.

**Fluorescent imaging of mitoDPP-3 with ML348/ML349 inhibitors**. In total 70,000–80,000 HEK293T cells/well were plated in 500 μL DMEM glutamax (10% FBS) into two wells of a four well chambered imaging dish (D35C4-20-1.5-N, Cellvis), which were precoated with 4 μg Poly-D-lysine (30–70 KDa, Alfa Aesar) for 2 h. After 32–48 h, the media was replaced by 1 μM Hoechst 33342, 100 nM MitoTracker Deep Red and DMSO/1 μM ML348/1 μM ML349 in 400 μL DMEM glutamax. After 30 min of incubation at 37 °C, the cells were washed with 400 μL of Live Cell Imaging Solution and replaced by 300 nM mitoDPP-3 with DMSO or 1 μM of ML348/ML349 in Live Cell Imaging Solution. After 10 min of incubation at 37 °C, images were obtained on an inverted epifluorescence microscope (Olympus IX83) attached with EMCCD camera (Photometrics Evolve Delta) with ×60 oil objective (N/A 1.42) for mitoDPP-3 (excitation filter 500/20, dichroic chroma 89007, emission filter 535/30, exposure time 200 ms, EM gain 75), MitoTracker Deep Red (excitation filter 640/30, emission filter 705/72, dichroic chroma 89016, exposure time 15 ms, EM gain 15), Hoechst 33342 (excitation filter 402/15, emission filter 455/50, dichroic chroma 89013, exposure time 40 ms, EM gain 20), and brightfield (exposure time 100 ms, EM gain 50). Analyses were performed in ImageJ (Wayne Rasband, NIH). For data analysis, the average fluorescence intensity per image in each experimental condition was obtained by gating on the MitoTracker signal and applying that to the corresponding mitoDPP-3 image, and the data were normalized to the average fluorescence intensity of the DMSO control. Each experiment was repeated in at least three biological replicates with identical results.

**Fluorescent imaging of mitoDPP-3 with APT1/APT2 shRNA**. 90,000 HEK293T cells/well were plated in 500 μL DMEM glutamax (10% FBS) into two wells of a four well chambered imaging dish (D35C4-20-1.5-N, Cellvis), which were precoated with 4 μg Poly-D-lysine (30–70 KDa, Alfa Aesar) for 2 h. After 18–20 h, the media was replaced with 500 μL DMEM glutamax (10% FBS) and the cells were transfected with 600 ng control/APT1/APT2 shRNAs following manufacture's conditions. Briefly, 26.2 μL of opti-MEM containing 1.8 μL of Lipofectamine 3000 was added to mix of 13.6 μL opti-MEM, 1.2 μL P3000 and 12 μL shRNAs mix (50 ng/μL), and resulting DNA:Lipofectamine mix was incubated at room temperature for 15 min. After incubation 52 μL of the DNA:Lipofectamine mix was added to the corresponding well of four well chambered imaging dish (D35C4-20-1.5-N, Cellvis). After 32–35 h the media was replaced by 1 μM Hoechst 33342 and 100 nM MitoTracker Deep Red in 400 μL DMEM glutamax. After 30 min of incubation at 37 °C, the cells were washed with 400 μL of Live Cell Imaging Solution and replaced by 300 nM mitoDPP-3 in Live Cell Imaging Solution. After 10 min of incubation at 37 °C, images were obtained as described above with the following settings: for mitoDPP-3 (exposure time 200 ms, EM gain 75), Mito-Tracker Deep Red (exposure time 15 ms, EM gain 15), Hoechst 33342 (exposure time 40 ms, EM gain 20), and brightfield (exposure time 100 ms, EM gain 50). Analyses were performed as described above and each experiment was repeated in at least three biological replicates with identical results.

**Immunofluorescence of APT1 localization**. Transient transfection was performed on cells seeded on glass coverslips in 6 well plates at about 60% confluency. 2 μg of plasmid and 6 μL of Fugene 6 (Promega) were added to each well using OptiMEM (GIBCO) as vehicle. siRNA-mediated gene silencing was performed using RNAi-MAX (Invitrogen). 0.03 nmol of siRNA and 4.5 μL of RNAiMAX were added to each well using OptiMEM (GIBCO) as vehicle. Cells were seeded on glass coverslips and transfected (or not) for 48 h (plasmids) or 72 h (siRNA). Cells were fixed with 3% paraformaldehyde for 20 min at 37 °C followed by three PBS washes, 10 min quenching in 50 mM NH₄Cl, three PBS washes, 5 min permeabilization with 0.1% TX100, three PBS washes and 0.5% BSA in PBS blocking overnight. Staining was performed with specific primary (anti-APT1/LYPLA1: abcam 91606, 1:500;

anti-TOM20: Santa Cruz sc-11415, 1:2000; anti-GM130: BD 610823, 1:1000; anti-c-MYC, homemade clone 9e10, 1:500) and secondary antibodies (Goat anti-mouse 488: Life Technologies A11029; Donkey anti-rabbit 568: Life Technologies A10042; Donkey anti-rabbit 647: Life Technologies A31573; Donkey anti-mouse 647: Life Technologies A31571) at room temperature for 30 min respectively including for each step three PBS washes. Finally, the coverslips were mounted in mowiol on glass slides and imaged with a LSM710 confocal microscope (Zeiss) with a ×63 oil immersion objective (NA 1.4). For denaturing conditions, the permeabilization with 0.1% TX100 was replaced by 30 min incubation with 5% SDS in 10 mM Tris base and 1 mM EDTA pH 9.0 followed by 5 min heating of the plate at 95 °C.

**Subcellular fractionation**. High quality mitochondria-enriched fractions were obtained using a continuous Percoll gradient-based protocol as described by Wieckowski et al.[63] Briefly, confluent HeLa cells were washed with ice-cold phosphate-buffered saline (PBS) without Ca(II) and Mg(II), collected by scraping and spun for 7 min at 600 g, 4 °C. Cell pellets were washed once with IB-1 (75 mM sucrose, 225 mM mannitol, 30 mM Tris–HCl, 0.1 mM EGTA, pH 7.4), re-spun for 7 min at 600 g, 4 °C, and finally resuspended in the right volume of IB-1. Cell homogenization was performed using a 7 mL glass-glass tight Dounce homogenizer (usually 10 strokes) and the nuclei and unbroken cells were spun down by two centrifugation steps of 7 min at 600 g. The resulting post-nuclear supernatant (PNS) was then aliquoted in 1.5-mL centrifuge tubes and spun for 10 min at 7000 g. The supernatant (SN1) was transferred to a new tube and the crude mitochondria pellet in IB-1 and both the pellet and supernatant were further spun for 10 min at 10,000 × g. The re-spun supernatant (SN1) was further centrifuged at 28,000 × g and 230,000 × g and the obtained pellets and final supernatant were kept for comparative western blot analysis. In turn, the washed crude mitochondria pellet was resuspended in mitochondria resuspension buffer (MRB, 250 mM mannitol, 5 mM HEPES, 0.5 mM EGTA, pH 7.4) and loaded on 10 mL of 30% Percoll solution (225 mM mannitol, 25 mM HEPES, 1 mM EGTA, 30% Percoll, pH 7.4) in an ultra-clear 14-mL polypropylene tube (Beckman 344060). The tube was filled until the top with MRB and gradients were ultracentrifuged for 35 min at 205,000 rcf at r max, 4 °C, using a Beckman SW40 rotor. Bands corresponding to mitochondria-associated ER membranes (MAMs) and mitochondria-enriched fractions were collected at the right positions in the tube and transferred to 2-mL centrifuge tubes. All samples were diluted 1:2 with MRB and spun twice for 10 min at 16,000 × g. The obtained pellets corresponding to heavy MAM and purified mitochondria were resuspended in MRB and stored at −80 °C until further analysis.

**SDS–PAGE and western blotting of mitochondrial fractionation**. To evaluate the quality of the fractions and the distribution of APT1 and other proteins, the protein content in all fractions was quantified using the BCA-based colorimetric assay. Around 10 μg of protein per fraction were loaded in pre-casted 4–20% gradient polyacrylamide gels (Invitrogen) and when the run was complete the gel was transferred on a nitrocellulose membrane using the iBlot gel transfer system (Invitrogen). Membrane blocking was achieved by 30 min incubation with PBS supplemented with 5% non-fat milk and 0.2% Tween-20 at room temperature. Incubation with primary antibodies (anti-APT1/LYPLA1: abcam 91606 1:500, anti-alpha tubulin: Sigma T5168 1:3000, anti-TOM20: Santa Cruz sc-11415 1:2000, anti-GM130: BD 610823 1:1000) was performed overnight at 4 °C and incubation with secondary antibodies (sheep anti-mouse IgG-HRP: GE NA931V, goat anti-rabbit IgG-HRP: Santa Cruz sc-2004, both 1:3000) was performed for 1 h at room temperature. After that, the membranes were washed 4–6 times with PBS/0.2% Tween-20 and the chemiluminescence signal was developed using the Super Signal West Dura solutions from Thermo Scientific and a Fusion Solo chemiluminescence imaging system.

**Epifluorescent imaging of mitoDPP-2 with palmitate**. 250,000–300,000 HEK293T cells/well were plated in 700 μL DMEM glutamax (10% FBS) into 2 wells of a 4 well chambered imaging dish (D35C4-20-1.5-N, Cellvis), which were pre-coated with 4 μg Poly-D-lysine (30–70 KDa, Alfa Aesar) for 2 h. After 20–24 h, the media was replaced by 500 μL DMEM glutamax. After 6 h of starvation, the cells were treated for another 6 h with 500 μL of 1% BSA ± 1 mM Palmitate made with DMEM glutamax. Then, the media was replaced by 1 μM Hoechst 33342 and 100 nM MitoTracker Deep Red in 400 μL DMEM glutamax containing 1% BSA ± 1 mM Palmitate. After 30 min of incubation at 37 °C, the cells were washed with 400 μL of Live Cell Imaging Solution and replaced by 1 μM DPP-2/500 nM mitoDPP-2 in Live Cell Imaging Solution (Molecular Probes). After 10 min of incubation at 37 °C, images were obtained as described above with the following settings: for DPP-2 (exposure time 150 or 250 ms, EM gain 100 or 150), mitoDPP-2 (exposure time 150 ms, EM gain 75), MitoTracker Deep Red (exposure time 15 ms, EM gain 15), Hoechst 33342 (exposure time 40 ms, EM gain 20), and brightfield (exposure time 100 ms, EM gain 50). Analyses were performed as described above and each experiment was repeated in at least three biological replicates with identical results.

**Fluorescent imaging of DPP-2/mitoDPP-2 with ACOT1/11 RNAi**. 140,000 HEK293T cells/well were plated in 500 μL DMEM glutamax (10% FBS) into two

wells of a 4 well chambered imaging dish (D35C4-20-1.5-N, Cellvis), which were precoated with 4 µg Poly-D-lysine (30–70 KDa, Alfa Aesar). After 18–20 h, the media was replaced by 500 µL DMEM glutamax (10% FBS) and the cells were transfected with 600 ng control/ACOT1/ACOT11 shRNAs using protocol described above. After 54–58 h the media was replaced by 1 µM Hoechst 33342 and 100 nM MitoTracker Deep Red in 400 µL DMEM glutamax (10% FBS). After 30 min of incubation at 37 °C, the cells were treated and imaged, as described above. Each experiment was repeated in at least three biological replicates with identical results.

**Real-time quantitative PCR**. Approximately 250,000 HEK293T cells/well were plated in 1200 µL DMEM glutamax (10% FBS) into 12 well dish (3512, Costar Corning). After 20–22 h, 350 µL of growth media was removed and cells were transfected with 950 ng of either control/targeted shRNA following manufacture's conditions. Briefly, 40 µL of opti-MEM containing 2.83 µL of Lipofectamine 3000 was added to mix of 21.4 µL opti-MEM, 1.9 µL P3000 and 19 µL shRNAs mix (50 ng/µL), and resulting DNA:Lipofectamine mix was incubated at room temperature for 13–15 min. After incubation 83 µL of the DNA:Lipofectamine mix was added to the corresponding well of 12 well dish (3512, Costar Corning). After 52–56 h total RNA was extracted from cells using RNeasy Plus Mini Kit (Qiagen) and followed by reverse transcription using PrimeScript RT Reagent kit (Clontech TaKaRa) on 500 ng RNA. The qPCR was done on 100 times diluted cDNA using FastStart Essential DNA Green Master (Roche) and GAPDH as the internal control (Primer 1: TGCACCACCAACTGCTTAGC; Primer 2: GGCATGGACTGTGGTCATGAG) on Light Cycler 96 real time PCR system (Roche). QPCR data was analyzed by comparative $C_T$ method[64]. Fold reduction in mRNA levels of targeted genes by corresponding shRNA treated samples in comparison to non-targeting vector is reported (Supplementary Table 3). In addition, the ratio of Ct values of targeted gene to GAPDH is presented for both control shRNA and targeted shRNA (Supplementary Table 3).

**Western blotting of knockdowns**. 250,000 HEK293T cells/well were plated in 1200 µL DMEM glutamax (10% FBS) in a 12 well dish (3512, Costar Corning). After 20–22 h, 350 µL of growth media was removed and the cells were transfected with 950 ng of either control/targeted shRNA following manufacture's conditions. Briefly, 40 µL of opti-MEM containing 2.83 µL of Lipofectamine 3000 was added to mix of 21.4 µL opti-MEM, 1.9 µL P3000 and 19 µL shRNAs mix (50 ng/µL), and resulting DNA:Lipofectamine mix was incubated at room temperature for 13–15 min. After incubation, 83 µL of the DNA:Lipofectamine mix was added to the corresponding well of the 12 well dish (3512, Costar Corning). After 48 h, all the cells were transferred to the 6 well dish (Denville Scientific Inc.) in 2 mL of DMEM glutamax (10% FBS). After an additional two days, the cells were washed with DPBS and 350 µL of RIPA lysis buffer (50 mM Tris, 150 mM NaCl, 0.5% deoxycholate, 0.1% SDS, 1.0% TritonX-100, pH 7.4) was added. The lysed cell solution was transferred to 2 mL tubes and vortexed for three times, 10 s each, and then incubated at 4 °C on rotator for 2 h. The protein concentration was measured by BCA assay (23225, Thermo Scientific). Proteins were precipitated by methanol: chloroform:H$_2$O (4:1:3) and spun down at 16,200 rcf, 4 °C for 15 min. The aqueous layer was removed, 1 mL of methanol was added, and the tube was slowly inverted 5 times to mix the methanol and chloroform layers. The precipitate thus obtained after spinning at 16,200 rcf, 4 °C for 15 min was air dried and dissolved in 1× loading buffer containing 32 mM DTT and made from Laemmli SDS sample buffer (4×, J63615, Alfa Aesar) and 4% SDS containg ABE buffer (50 mM HEPES, 5 mM EDTA, 150 mM NaCl, pH 7.4) such that final protein concentration is 2.5 µg/µL by short sonication followed by heating at 95 °C for 5 min. 10 µL of this protein lysate (25 µg) was then loaded onto a 10% SDS–PAGE gel. The proteins were then transferred onto PVDF membrane (IPVH00010, pore size 0.45 µm, Immobilon-P, Milipore) that was pre-activated with methanol for 5 min with transfer buffer (25 mM Tris, pH 8.3, 190 mM glycine, and 20% MeOH) at 100 V for 120 min. After transfer, the membrane was blocked for 1 h with 5% BSA (BP9703, Fischer) solution in wash buffer (20 mM Tris, pH 7.5, 150 mM NaCl, 0.1% Tween-20), followed by overnight incubation with either 1:4000 rabbit polyclonal anti-calnexin (abcam, ab22595), 1:1000 mouse monoclonal anti-ACOT1 (sc-373919, SCBT) or 1:1000 rabbit monoclonal anti-APT1 (ab91606, abcam) in 5% BSA solution in wash buffer. The membrane was washed with 5 × 10 mL of ice cold wash buffer for 5 min followed by 1 h incubation with either 1:2000 goat anti-rabbit IgG HRP (sc-2004 SCBT) or 1:2000 goat anti-mouse IgG HRP (sc-2005 SCBT) in 5% BSA solution in wash buffer, washed for 5 min with 5 × 10 mL of ice cold wash buffer, and then visualized using Super Signal West Pico Plus (34577, Thermo Scientific) recorded on Fluor Chem R (Protein Simple) imaging station.

**Data availability**. All data generated or analyzed during this study are included in the published article (and its supplementary information) or are available from the corresponding authors on reasonable request.

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

## Acknowledgements

This work was supported by the University of Chicago, the National Institute of General Medical Sciences of the National Institutes of Health (R35 GM119840) to B.C.D., the National Cancer Institute (RO1 CA200310) to KFM, the University of Chicago Medicine Comprehensive Cancer Center (P30 CA14599), a "Catalyst Award" to B.C.D. from the Chicago Biomedical Consortium, with support from the Searle Funds at The Chicago Community Trust, a Research Fellowship from the Alfred P. Sloan Foundation to B.C.D., the EPFL and the Swiss National Science Foundation to F.G.v.d.G. The research leading to these results has also received funding from the European Research Council under the European Union's Seventh Framework Programme (FP/2007–2013)/ERC Grant Agreement No. 340260—PalmERa'. M.E.Z. was a recipient of a long-term post-doctoral fellowship from the Human Frontier Science Program (LT000152/2014-L) and a transition postdoctoral fellowship from the Swiss SystemsX.ch initiative (TPdF 2013/143), evaluated by the Swiss National Science Foundation. We thank C. He (University of Chicago), A. Mukherjee (National Institutes of Health), and Y. Krishnan (University of Chicago) for supplying materials and equipment, and P. Bastiaens (Max Planck Institute Dortmund) for providing the APT1/2-mCitrine constructs.

## Author contributions

R.S.K. and P.D.E. synthesized all compounds. R.S.K. performed all analytical measurements, in vitro assays, and cell culture experiments. Y.C. measured enzymatic kinetic parameters. P.A.S., M.-E.Z., and F.G.v.d.G. performed immunofluorescence and biochemical APT1 localization studies. M.Z.S., L.E.D., and K.F.M. prepared live mitochondria. R.S.K. and B.C.D. designed experimental strategies and wrote the paper with input from all authors.

## Additional information

**Competing interests:** B.C.D. and R.S.K. have filed a provisional patent on the DPPs. The remaining authors declare no competing financial interests.

