## [Peer Review File · Nature Communications]

Reviewers' comments:

Reviewer #1 (Remarks to the Author):

The manuscript by Kathayat R et al describes the synthesis and evaluation of mitochondrial targeted S-deacylation probes in vitro and in mammalian cells. This work is an extension of previous studies from the Dickinson lab on their original fluorogenic S-deacylation probes. By modifying DPPs with TPP group, previously described in other imaging studies, the authors generated mito-DPPs. The synthesis and characterization of mito-DPPs are well described in SI and also showed fluorogenic activity with protein S-deacylase in vitro. In mammalian cells, mito-DPPs localized to mitochondria nicely, as judged by contain with mitotracker. The author then demonstrate mito-DPP fluorescence imaging is sensitive to protein S-deacylase inhibitors and siRNA knockdown of APT1, in particular. Furthermore, the mito-DPP fluorescence is also increased by 1 mM palmitate and selective siRNA knockdown of ACOT1 and 11, suggesting metabolic and local enzymatic perturbations of palm-CoA levels affects protein S-deacylase activity in cells. Notably, siRNA knockdown of ACOT1 and 11 selectively modulated cytosolic DPP2/3 activity versus mito-DPP2/3.

This is a well executed and described study that warrants publication in Nat Comm, if that authors can address several key points listed below.

1) Would be great for authors to include experiments with non-targeted DPP2/3 and non-hydrolyzable analog with APT1 knockdown.

2) activity of DPP2/3 and mito-DPP2/3 with purified mitochondrial in vitro?

3) activity of mito-DPP2/3 with overexpression of mitochondria targeted APT1?

4) validate and report levels of siRNAi knockdown of target genes/proteins.

5) Further discussion is need to describe how elevated levels of palmitoyl-CoA increase DPP activity. One could envision the opposite result where increase in palmitoyl-CoA elevates S-palmitoylation and decreases DPP activity. The authors should discuss these possibilities and the interpretation of their data more precisely.

Reviewer #2 (Remarks to the Author):

Kathayat et al. describe the synthesis and use of a mitochondrial targeted S-deacylase probe for the quantification of overall mitochondrial S-deacylase activity. This work extends their previous work with cytosolic deacylase fluorescent probes. The authors thoroughly show that the probe is able to translocate to mitochondria and is acted on enzymatically within the organelle. The authors also demonstrate that their probe is sensitive to biological perturbations including shRNA treatment of putative mitochondrial deacylases and palmitate treatment. The probes presented here have high potential to explore the effects of S-deacylation in mitochondria, however this real utility needs to be shown more convincingly before the manuscript might be ready for publication in Nature Communications.

The following are my major critiques:

1. The mitochondrial localization of APT1 is still ambiguous. The Western blot in Supplementary Figure 5 does not show a strong enrichment of mitochondria (Hsp60). The authors could perform mitochondrial localization studies using immunohistochemistry on native or tagged APT1 or performing more rigorous mitochondrial enrichment experiments. Does APT1 have a mitochondrial targeting sequence?

2. The effect of APT1 shRNA on mitoDPP-2 fluorescence is statistically significant, but the effect

size is small. The y-axis scale in Figure 4b is misleading, and it should begin at 0 and have a break until 0.7 so as to not mask the modest effect of APT1 shRNA on mitoDPP-2 fluorescence.

3. The authors demonstrate that knockdown of APT1, APT2, ACOT1, and ACOT11 have varying effects on probe fluorescence. The percentage knockdown should be quantified to verify that the transcript abundance is down or perform Western blots to demonstrate that protein abundance is decreased during treatment.

4. The authors claim that mitoDPP-2 is a "pan-activity" deacylase probe for mitochondria yet are unable to identify all mitochondrial deacylases. Could some mitochondrial S-deacylases prefer the octanoate handle whereas others do not, making the probe biased towards certain S-deacylases? Are there other mitochondrial proteins besides known S-deacylases that could perform the chemistry necessary to activate the probe?

5. Are the IC50s for the inhibitors ML348 and ML349 different such that treating with the same amount of inhibitor (1 μ M) would have an effect on mitochondrial APT1 but not mitochondrial APT2?

The following are minor points:

1. Is there an upper limit on the fluorescence of the probe? Would adding more of the probe to cells yield a larger dynamic range?
2. The pH of the mitochondrial matrix is different than that of cytosol. Is the probe sensitive to physiological pH changes?
3. What is the K_m of the probes for APT1 and APT2 and how could that affect the in vivo fluorescence?
4. Does octanoate released from the probe have an effect on mitochondrial metabolism?

Note for major point 2: The authors state that "Based on the genetic screen, APT1 accounts for a significant fraction of the total mitochondrial S-deacylase activity, but not all of it." Knockdown of APT1 only accounts for ~15% of deacylase activity, which does not appear to be a significant fraction.

Reviewer #3 (Remarks to the Author):

This MS builds on the previous invention of the Dickinson group, published in *Nat. Chem. Biol.* (2017) concerning fluorescent probes (DPP-1-3) that allow investigations on cysteine depalmitoylation in living cells. Here they describe the development of mitochondrial-targeted DPPs through the introduction of a triphenylphosphonium moiety, and apply the novel probes to obtain evidence for enzymes involved in cysteine depalmitoylation within mitochondria. Publication of this MS in *Nature Communications* is not recommended, as there are major technical shortcomings which significantly weaken the drawn conclusions. The reasons behind this statement are set out below. If the technical issues can be resolved then publication in a more specialized journal may be warranted.

In vitro, mitoDPP-2 works as-designed, but there is no novelty compared to their previous report. The biochemical characterization of this probe does not reflect mitochondrial or cytosolic pH and salt, which may significantly alter mitoDPP-2 hydrolysis by APT-1/2 and the fluorescence emission of the hydrolyzed dye.

The authors show by confocal microscopy that mitoDPP-2 is fluorescent in living cells and that its signal co-localizes with MitoTracker, indicating a mitochondrial accumulation of deacylated probe. The (mitochondrial) hydrolysis of mitoDPP-2 could be reduced to a large extent (~80%), by treatment with Palm B, and the authors also show that a mitochondria-enriched fraction of MCF7

cells cleaves DPP1, which could also be blocked by Palm B. Together these data indicate that hydrolysis is indeed mediated by enzymatic activity and not spontaneous hydrolysis, as the authors state. However, it is not clear if mitoDPP-2 hydrolysis takes place within the mitochondria, or whether the dye simply translocates and accumulates into the mitochondrial compartment after hydrolytic cleavage in the cytosol. This is not unlikely, since the expression of APT1 is magnitudes higher in the cytosol compared to mitochondria (Supplemental Figure 5). Control experiments should be performed with deacylated probe in living cells to reveal whether it translocates in a similar fashion. In addition, confirmation of cysteine depalmitoylation is shown by applying DPP-1 which functions via an identical *modus operandi*, arguably suffering from similar artefacts. Depalmitoylation should be confirmed with an unrelated technique, such as metabolic labelling of palmitoylated proteins with alkyne or azide-tagged palmitate.

Next, the authors went on to identify the enzyme(s) responsible for mitoDPP-2 hydrolytic activity by genetic knockdown (KD) of several known depalmitoylases (APT1, APT2, PPT2, PPT1, PPT2, LYPLAL and ABHD17A-C). They used an assay in 96-well format and screened for changes in mitoDPP-2 hydrolysis. However, no evidence is provided that mitoDPP-2 can in fact be cleaved by PPT2, PPT1, PPT2, LYPLAL and ABHD17A-C, similar to the biochemical experiments shown in Figure 2. In addition, the authors do not present evidence of KD efficiency such as by monitoring mRNA levels and showing Western blots for the different depalmitoylases, and analyzing potential compensatory mechanisms.

The authors claim that APT1 KD significantly reduces mitoDPP-2 hydrolysis. A small effect of APT1 KD is indeed observed ~14%. However, the error bars (SEM) in the bar graph (Figure 4) are large and the statistical power to support a significant role of APT1 is weak. Additional replicates might more convincingly show a significant effect of APT1 KD on mitoDPP-2 hydrolysis. These extra replicates could have been easily performed in the 96-well assay format. Scaling of the bar graph in Figure 4B from 0 to 1.2 would have given a more representative impression of the results than the scale from 0.7 to 1.2 chosen by the authors, which clearly emphasizes what is in fact a small change. It is important to note that the remaining 86% of mitoDPP-2 signal is mediated by enzymes other than APT1, and this should have been discussed by the authors. The addition of the effect of Palm B in Figure 4B would be informative.

Minimal parts of the Western blots (Supporting Figure 5) are shown to demonstrate APT1 expression in mitochondria. Here the authors must show complete blots. The expression of APT1 in mitochondria seems very low compared to the cytosol, and their data point towards other enzyme(s) playing a much more prominent role in (mitochondrial) hydrolysis of mitoDPP2 and a minimal role of APT1. The question that remains is to what degree is APT1 outside the mitochondria responsible for the fluorescent dye visible within the mitochondria?

As a next step, the authors develop a probe more specific for APT-1 (mitoDPP-3). Again, the conditions used for biochemical characterization shown in Figure 5 do not reflect mitochondrial and cytosolic conditions. Feeding mitoDPP-3 to cells resulted again in co-localization with MitoTracker and fluorescence of the hydrolyzed dye was reduced by Palm B, ML348 and by KD of APT1 as shown by microscopy data shown in Supplementary Figures 7-10. However, the images are highly heterogeneous, with cells only containing MitoTracker, only mitoDPP-3 and cells containing both. It is unclear how the authors proceeded to reliably quantify the effect of the inhibitors and KD, which should have been analyzed by FACS analysis. Again, the authors fail to show KD efficiency with appropriate mRNA levels and Western blots.

Finally, the authors went on to identify ACOT1 and ACOT11 as mediators of mitochondrial cysteine palmitoylation. The statistical power of Figure 8A and B is again low and more replicates are necessary to convincingly show a significant effect of ACOT1 and ACOT11 KD on the DPP probes in the cytosol and mitochondria respectively. In addition, again, the authors should monitor KD efficiency by measuring mRNA levels and by appropriate Western blots.

As minor corrections, the authors should make a clear distinction between 'deacylation' and 'depalmitoylation' as their substrate lipid is C8; and the sequences used for knockdown (Supplementary Table 1) are not shRNA.

Reviewer #1 (Remarks to the Author):

The manuscript by Kathayat R et al describes the synthesis and evaluation of mitochondrial targeted S-deacylation probes *in vitro* and in mammalian cells. This work is an extension of previous studies from the Dickinson lab on their original fluorogenic S-deacylation probes. By modifying DPPs with TPP group, previously described in other imaging studies, the authors generated mito-DPPs. The synthesis and characterization of mito-DPPs are well described in SI and also showed fluorogenic activity with protein S-deacylase *in vitro*. In mammalian cells, mito-DPPs localized to mitochondria nicely, as judged by contain with mitotracker. The author then demonstrate mito-DPP fluorescence imaging is sensitive to protein S-deacylase inhibitors and siRNA knockdown of APT1, in particular. Furthermore, the mito-DPP fluorescence is also increased by 1 mM palmitate and selective siRNA knockdown of ACOT1 and 11, suggesting metabolic and local enzymatic perturbations of palm-CoA levels affects protein S-deacylase activity in cells. Notably, siRNA knockdown of ACOT1 and 11 selectively modulated cytosolic DPP2/3 activity versus mito-DPP2/3. This is a well-executed and described study that warrants publication in Nat Comm, if that authors can address several key points listed below.

We would like to thank reviewer for the positive comments on our work.

1) Would be great for authors to include experiments with non-targeted DPP2/3 and non-hydrolyzable analog with APT1 knockdown.

As requested, and due to additional new experiments that are no added, we have included a variety of experiments with DPP-2 and/or DPP-3 throughout the paper (see Fig. 9, Supp. Fig. 12, etc). We cannot use a non-hydrolyzable analog of the DPPs because the background of the probe is so low that we would see no signal (the “off” form of the DPPs is essentially non-fluorescent).

2) activity of DPP2/3 and mito-DPP2/3 with purified mitochondrial *in vitro*?

As suggested, we have performed *in vitro* assays with purified live mitochondria and replaced the previous lysate experiments with this significantly better experiment (see Supp. Fig. 6). We thank reviewer for this suggestion which improved the manuscript.

3) activity of mito-DPP2/3 with overexpression of mitochondria targeted APT1?

We thought this was an excellent proposed experiment. Therefore, we generated and APT1 expression vector with a Cox8 mitochondrial targeting sequence as a tool to overexpress an APT1 in the mitochondria and see how our probes performed. The results of this experiment were so striking that we decided to add a new main text fig (Fig. 4). In short, over-expressing an APT1 in the mitochondria has no effect on the cytosolic DPP-2 signal, but causes a dramatic increase in the mitoDPP-2, mitochondrial signal. The original version of the manuscript was lacking the crystal clear evidence that the mitoDPPs were measuring mitochondrial enzymatic activity (as brought up by all three reviewers), which this experiment now provides. Thanks for the suggestion! Moreover, as seen in the new manuscript, we found that APT1 is actually already naturally localized predominantly in mitochondria, so this experiment was actually in retrospect gratuitous, but still a nice control.

4) validate and report levels of siRNAi knockdown of target genes/proteins.

We performed RT-qPCR for all of the genes in our APT knockdown screen (Supp. Table 3) and also further validated the knockdown of APT1 and ACOT1 by Western blot (Supp. Fig. 16). Unfortunately, we made multiple attempts at measuring ACOT11 by Western blot, using two different antibodies from two different vendors, but failed to get adequate results. The antibodies for this relatively unstudied gene are not satisfactory, but fortunately the RT-qPCR data for this gene (performed in replicates) showed clearly the gene was knocked down.

5) Further discussion is need to describe how elevated levels of palmitoyl-CoA increase DPP activity. One could envision the opposite result where increase in palmitoyl-CoA elevates S-palmitoylation and decreases DPP activity. The authors should discuss these possibilities and the interpretation of their data more precisely.

We appreciate these thoughts and the possibility that palmitoyl-CoA deactivates S-depalmitoylases in order to increase overall S-palmitoylation level. This final experiment was really a hypothesis-generating experiment, as we did not know what to expect. One could rationalize it both ways, and we added additional discussion on these points. We are interested in now pursuing what the role of this change is, and critically, how targets are changed in their lipidation levels accordingly. Our hope is that this result encourages others, especially proteomics groups, to get interested in the mitochondria as well.

Reviewer #2 (Remarks to the Author):

Kathayat et al. describe the synthesis and use of a mitochondrial targeted S-deacylase probe for the quantification of overall mitochondrial S-deacylase activity. This work extends their previous work with cytosolic deacylase fluorescent probes. The authors thoroughly show that the probe is able to translocate to mitochondria and is acted on enzymatically within the organelle. The authors also demonstrate that their probe is sensitive to biological perturbations including shRNA treatment of putative mitochondrial deacylases and palmitate treatment. The probes presented here have high potential to explore the effects of S-deacylation in mitochondria, however this real utility needs to be shown more convincingly before the manuscript might be ready for publication in Nature Communications.

The following are my major critiques:

1. The mitochondrial localization of APT1 is still ambiguous. The Western blot in Supplementary Figure 5 does not show a strong enrichment of mitochondria (Hsp60). The authors could perform mitochondrial localization studies using immunohistochemistry on native or tagged APT1 or performing more rigorous mitochondrial enrichment experiments. Does APT1 have a mitochondrial targeting sequence?

We agree with the reviewer that the data in our original submission was inadequate. Therefore, we formed a collaboration with Gisou van der Goot (EPFL), who is now a coauthor on the manuscript, to test these exact points. In short, we found that the literature was incorrect about APT1 localization. In fact, we found that APT1 expression, or, now for the first time, endogenous APT1, is primarily localized in the mitochondria! The reason the previous studies got this wrong is that fusion to large fluorescent proteins perturbs APT1 localization (but interestingly, not APT2 localization). Additionally, we performed much more careful and precise fractionation experiments, as suggested, and confirmed that APT1 is indeed primarily in the mitochondrial fractions (Fig. 7c). We added an entire new section to the paper discussing these results, along with a new main text figure (Fig. 7) and corresponding Supplementary Figures.

2. The effect of APT1 shRNA on mitoDPP-2 fluorescence is statistically significant, but the effect size is small. The y-axis scale in Figure 4b is misleading, and it should begin at 0 and have a break until 0.7 so as to not mask the modest effect of APT1 shRNA on mitoDPP-2 fluorescence.

We apologize for presenting the data in this way. The old figures showing the screen were performed on a plate reader assay, which has a very high background, and therefore does not show dramatic differences. The purpose of these experiments were just to find initial putative hits, which we would then validate by very careful microscopy. We never meant for the

screening assays to be confirmative, so we moved those figures to the SI and focused the main text figures on the imaging experiments. Also, as requested, we adjusted the y-axis on all figures to be 0.

3. The authors demonstrate that knockdown of APT1, APT2, ACOT1, and ACOT11 have varying effects on probe fluorescence. The percentage knockdown should be quantified to verify that the transcript abundance is down or perform Western blots to demonstrate that protein abundance is decreased during treatment.

This was also requested by reviewer 1. We performed RT-qPCR for all of the genes in our APT knockdown screen (Supp. Table 3) and also further validated the knockdown of APT1 and ACOT1 by Western blot (Supp. Fig. 16). Unfortunately, we made multiple attempts at measuring ACOT11 by Western blot, using two different antibodies from two different vendors, but failed to get adequate results. The antibodies for this relatively unstudied gene are not satisfactory, but fortunately the RT-qPCR data for this gene (performed in replicates) showed clearly the gene was knocked down.

4. The authors claim that mitoDPP-2 is a “pan-activity” deacylase probe for mitochondria yet are unable to identify all mitochondrial deacylases. Could some mitochondrial S-deacylases prefer the octanoate handle whereas others do not, making the probe biased towards certain S-deacylases? Are there other mitochondrial proteins besides known S-deacylases that could perform the chemistry necessary to activate the probe?

This is a very good point brought up by the reviewer, and something we think about a lot. The reality is no probe is perfect, so genetic controls are always needed to confirm a measured activity. Therefore, we cannot say with 100% confidence that all of the signal we are measuring is due to S-depalmitoylases. That being said, our probe is a peptide with a thioester, so it's as native as possible, aside from the surrogate lipid. S-palmitoylation is by far the dominant modification, and we have found no evidence S-octonylation is biologically-relevant. Therefore, we believe that much of the activity measured by our probes is relevant to de-palmitoylation. Future genetic screens (ongoing) will tell us how true this is. We added additional text addressed these potential concerns in the discussion to point out potential false-positives with the DPPs.

5. Are the IC50s for the inhibitors ML348 and ML349 different such that treating with the same amount of inhibitor (1 μ M) would have an effect on mitochondrial APT1 but not mitochondrial APT2?

The APT2 inhibitor (ML349) is actually a slightly better inhibitor than ML348, so probably we are okay. In any case, none of these experiments are meant to be exclusionary, and we are careful to never state that APT2 for example is not ever present in mitochondria. The data here (imaging and localization studies) just does not provide any evidence for APT2 in the mitochondria.

The following are minor points:

1. Is there an upper limit on the fluorescence of the probe? Would adding more of the probe to cells yield a larger dynamic range?

We see no inhibition by the probes, so essentially the probes get processed at some rate and we take a snapshot of that reaction at whatever time we do the imaging at (typically 10-20 min). We usually use the lowest concentration of probe possible to avoid perturbing the system. However, yes, as you add more probe you get more signal per time, as expected. The DPP scaffold used here is a turn-on approach, which gives really a binary answer (are the cells brighter or dimmer than another set of cells). The advantage is the probes are very sensitive and easy to use.

However, we also recently developed a ratiometric scaffold (the RDPs, <http://pubs.rsc.org/en/content/articlelanding/2017/sc/c7sc02805a#!divAbstract>), which allows us to better quantitate the reaction completion. Although the scaffolds are different, the *in vitro* kinetics are similar, so we can assume the reaction completion numbers are comparable. One next step might be to make mitoRDPs, which would then allow us to perform much more quantitative experiments in the mitochondria. However, again the ease-of-use and sensitivity of the mitoDPPs make them more useful for the majority of cell culture experiments.

2. The pH of the mitochondrial matrix is different than that of cytosol. Is the probe sensitive to physiological pH changes?

This is a great point. We performed additional kinetic experiments under more mitochondrial-relevant conditions (new Supp. Fig. 2). The probe still performed great. Rhodols (like the mitoDPPs) are sensitive to acidic conditions, so late endosomes would be a bad place to try and image. But alkaline pH has minimal effects on the fluorescence.

3. What is the K_m of the probes for APT1 and APT2 and how could that affect the *in vivo* fluorescence?

We performed kinetic measurements of mitoDPP-2 and mitoDPP-3 with APT-1 (new Supp. Table 1 and Supp. Fig. 1). We are not sure exactly where the selectivity comes from in cells. We have no evidence that APT2 is in mitochondria, so this is not the correct “off-target” comparison.

Once we identify other mitochondrial APTs our hypothesis is that they will not prefer a lysine at the C-terminal position of the substrate, but that remains to be seen.

4. Does octanoate released from the probe have an effect on mitochondrial metabolism?

This is an interesting question. At the concentrations we are using (100s of nM) and the fact that the reaction is not complete during imaging, we are generating low (10s-100s of nM) octanoate. Also, these experiments only take 10-20 min, so they are not sitting around too long. However, as with any probe, one has to worry about perturbing the system. In this case, we see no evidence of toxicity with the probes. This is a concern with any small molecule approach, but given the very low concentrations and short experiments needed here, does not give me any pause.

Note for major point 2: The authors state that “Based on the genetic screen, APT1 accounts for a significant fraction of the total mitochondrial S-deacylase activity, but not all of it.” Knockdown of APT1 only accounts for ~15% of deacylase activity, which does not appear to be a significant fraction.

We apologize for this poor wording. “Significant fraction” is definitely not what the data shows. We reworded this section accordingly.

Reviewer #3 (Remarks to the Author):

This MS builds on the previous invention of the Dickinson group, published in Nat. Chem. Biol. (2017) concerning fluorescent probes (DPP-1-3) that allow investigations on cysteine depalmitoylation in living cells. Here they describe the development of mitochondrial-targeted DPPs through the introduction of a triphenylphosphonium moiety, and apply the novel probes to obtain evidence for enzymes involved in cysteine depalmitoylation within mitochondria. Publication of this MS in Nature Communications is not recommended, as there are major technical shortcomings which significantly weaken the drawn conclusions. The reasons behind this statement are set out below. If the technical issues can be resolved then publication in a more specialized journal may be warranted.

We thank the reviewer for their careful reading of the manuscript. We believe the revised version, which features additional controls and conclusions, corrects the noted short-comings and is now suitable for Nat. Commun.

In vitro, mitoDPP-2 works as-designed, but there is no novelty compared to their previous report. The biochemical characterization of this probe does not reflect mitochondrial or cytosolic pH and salt, which may significantly alter mitoDPP-2 hydrolysis by APT-1/2 and the fluorescence emission of the hydrolyzed dye.

We agree that the mechanism of activation of the mitoDPPs is identical to that of the DPPs. However, successfully delivering a small molecule to the mitochondria is not trivial, and required substantial optimization. None-the-less, we believe the real novelty is the biological discoveries – active APTs in mitochondria, APT1 is primarily in mitochondria, and the APTs are dynamic. Reviewer 2 also requested we perform additional *in vitro* experiments under more mitochondrial conditions, which we did and included as new Supp. Fig. 2.

The authors show by confocal microscopy that mitoDPP-2 is fluorescent in living cells and that its signal co-localizes with MitoTracker, indicating a mitochondrial accumulation of deacylated probe. The (mitochondrial) hydrolysis of mitoDPP-2 could be reduced to a large extent (~80%), by treatment with Palm B, and the authors also show that a mitochondria-enriched fraction of MCF7 cells cleaves DPP1, which could also be blocked by Palm B. Together these data indicate that hydrolysis is indeed mediated by enzymatic activity and not spontaneous hydrolysis, as the authors state. However, it is not clear if mitoDPP-2 hydrolysis takes place within the mitochondria, or whether the dye simply translocates and accumulates into the mitochondrial compartment after hydrolytic cleavage in the cytosol. This is not unlikely, since the expression of APT1 is magnitudes higher in the cytosol compared to mitochondria (Supplemental Figure 5). Control experiments should be performed with deacylated probe in living cells to reveal whether it translocates in a similar fashion. In addition, confirmation of cysteine depalmitoylation is shown by applying DPP-1 which functions via an identical *modus operandi*, arguably suffering from similar artefacts. Depalmitoylation should be confirmed with an unrelated technique, such as metabolic labelling of palmitoylated proteins with alkyne or azide-tagged palmitate.

The other reviewers brought up similar concerns. First off, we tried the experiment as recommended (using the deacylated probe). This was inconclusive as the deacylated probe is not very cell permeable at all (which is actually evidence of localization prior to reaction). However, Reviewer 1 suggested a great experiment that ended up working well to address this exact concern. We generated a mitochondrial-targeted APT, which we found only caused a change in the mitoDPP, and not a DPP. This is confirmation that the mitoDPPs are sensitive to mitochondrial enzymes. The results of this experiment were so striking that we decided to add a new main text fig (Fig. 4). Finally, we discovered that the literature is incorrect, and APT1 is in fact primarily localized in mitochondria. The reason the previous studies got this wrong is that

fusions to fluorescent proteins perturbs APT1 localization (but interestingly, not APT2 localization). Additionally, we performed much more careful and precise fractionation experiments and confirmed that APT1 is indeed primarily in the mitochondrial fractions (Fig. 7c). We added an entire new section to the paper discussing these results, along with a new main text figure (Fig. 7) and corresponding Supplementary Figures. We believe proteomic assays to identify mitochondrial targets of APTs is outside of the scope of this work. As we discuss in the paper, these experiments are hampered by an inability to selectively perturb mitochondrial APTs. We are working on targeted inhibitors and other genetic tools for this exact reason, but again, that work is outside of the scope of this paper and the conclusions we wish to present here.

Next, the authors went on to identify the enzyme(s) responsible for mitoDPP-2 hydrolytic activity by genetic knockdown (KD) of several known depalmitoylases (APT1, APT2, PPT2, PPT1, PPT2, LYPLAL and ABHD17A-C). They used an assay in 96-well format and screened for changes in mitoDPP-2 hydrolysis. However, no evidence is provided that mitoDPP-2 can in fact be cleaved by PPT2, PPT1, PPT2, LYPLAL and ABHD17A-C, similar to the biochemical experiments shown in Figure 2. In addition, the authors do not present evidence of KD efficiency such as by monitoring mRNA levels and showing Western blots for the different depalmitoylases, and analyzing potential compensatory mechanisms.

Similar requests were made by other reviewers. We performed RT-qPCR for all of the genes in our APT knockdown screen (Supp. Table 3) and also further validated the knockdown of APT1 and ACOT1 by Western blot (Supp. Fig. 16). Unfortunately, we made multiple attempts at measuring ACOT11 by Western blot, using two different antibodies from two different vendors, but failed to get adequate results. The antibodies for this relatively unstudied gene are not satisfactory, but fortunately the RT-qPCR data for this gene (performed in replicates) showed clearly the gene was knocked down. We do not mean to state that the mitoDPPs are substrates for the enzymes in our screen – just that each are part of the family. In fact, there is very strong evidence that PPT2 and LYPLAL1, for example, are definitely not S-depalmitoylases. The data and our conclusions are not exclusionary of the genes tested, but the lack of activity is evidence that they are not active in mitochondria on the mitoDPPs (i.e. a S-acyl peptide substrate).

The authors claim that APT1 KD significantly reduces mitoDPP-2 hydrolysis. A small effect of APT1 KD is indeed observed ~14%. However, the error bars (SEM) in the bar graph (Figure 4) are large and the statistical power to support a significant role of APT1 is weak. Additional replicates might more convincingly show a significant effect of APT1 KD on mitoDPP-2 hydrolysis. These extra replicates could have been easily performed in the 96-well assay format. Scaling of the bar graph in Figure 4B from 0 to 1.2 would have given a more representative

impression of the results than the scale from 0.7 to 1.2 chosen by the authors, which clearly emphasizes what is in fact a small change. It is important to note that the remaining 86% of mitoDPP-2 signal is mediated by enzymes other than APT1, and this should have been discussed by the authors. The addition of the effect of Palm B in Figure 4B would be informative.

A similar concern was brought up by other reviewers. The problem was that we presented the plate reader assay, which has a very high error and limited dynamic range compared to imaging, in the main text. This experiment was never meant to be conclusive, but simply to guide more careful imaging experiments. We therefore moved these plate reader assays to the SI (and readjusted the axis as requested). We also added new flow cytometry data (new Supp. Fig. 12) confirming the plate reader data. The main results are now shown in the updated main text figures, which show that APT1 inhibition with ML348 causes a ~50% reduction in mitoDPP-3 and shRNA knockdown causes a ~20% reduction. Indeed, for mitoDPP-2 the fraction is less, which does suggest other enzymes can process the mitoDPPs. As requested, we added additional text about this point. We are very actively looking for other enzymes in the mitochondria that can do this chemistry, and look forward to reporting those results in the near future.

Minimal parts of the Western blots (Supporting Figure 5) are shown to demonstrate APT1 expression in mitochondria. Here the authors must show complete blots. The expression of APT1 in mitochondria seems very low compared to the cytosol, and their data point towards other enzyme(s) playing a much more prominent role in (mitochondrial) hydrolysis of mitoDPP2 and a minimal role of APT1. The question that remains is to what degree is APT1 outside the mitochondria responsible for the fluorescent dye visible within the mitochondria?

As mentioned above, we repeated those fractionation experiments much more carefully in collaboration with Gisou van der Goot (EPFL), who is now a coauthor on the paper. We also performed a series of careful imaging experiments (new Fig. 7). In short, the literature is incorrect about APT1 localization – it is primarily in the mitochondria. We think this result not only validates the mechanism of the mitoDPPs, but is an excellent demonstration of the power of small molecules to uncover spatially-regulated biochemical signaling.

As a next step, the authors develop a probe more specific for APT-1 (mitoDPP-3). Again, the conditions used for biochemical characterization shown in Figure 5 do not reflect mitochondrial and cytosolic conditions. Feeding mitoDPP-3 to cells resulted again in co-localization with MitoTracker and fluorescence of the hydrolyzed dye was reduced by Palm B, ML348 and by KD of APT1 as shown by microscopy data shown in Supplementary Figures 7-10. However, the images are highly heterogeneous, with cells only containing MitoTracker, only mitoDPP-3 and

cells containing both. It is unclear how the authors proceeded to reliably quantify the effect of the inhibitors and KD, which should have been analyzed by FACS analysis. Again, the authors fail to show KD efficiency with appropriate mRNA levels and Western blots.

As described above, we now include confirmation of all relevant KDs. We also included FACS analysis for the key result – APT1 activity (new Supp. Fig. 12). I would disagree that the images are too heterogeneous to quantify – mitochondria are inherently heterogeneous (see the mitotracker signals). We perform all experiments multiple times on different days and are very careful about our quantification. Additionally, as a lab rule, only “average” images are allowed into papers (not exceptional images), which may be one reason why the example images are not as striking as the quantification. Our goal is that if someone repeats our experiments that they will get results as good or better than those presented in the paper, which is of course not always the case with new probes. We also go through great lengths to add details about how all images were conducted and analyzed in the Methods. We are very confident in all of the results presented in this paper. Molecular imaging, if conducted carefully, can be a very informative approach.

Finally, the authors went on to identify ACOT1 and ACOT11 as mediators of mitochondrial cysteine palmitoylation. The statistical power of Figure 8A and B is again low and more replicates are necessary to convincingly show a significant effect of ACOT1 and ACOT11 KD on the DPP probes in the cytosol and mitochondria respectively. In addition, again, the authors should monitor KD efficiency by measuring mRNA levels and by appropriate Western blots.

As described above, we now include confirmation of all relevant KDs. Also, as with the other plate reader assays, these experiments were always as a guide to find interesting targets, not as confirmation. Therefore, we moved these figures to the SI so the main text could focus in on the molecular imaging results (new Fig. 9), which shows significant effects of ACOT1 and ACOT11 KD (25-50%) on the cytosolic and mitochondrial DPPs, respectively.

As minor corrections, the authors should make a clear distinction between ‘deacylation’ and ‘depalmitoylation’ as their substrate lipid is C8; and the sequences used for knockdown (Supplementary Table 1) are not shRNA.

We agree with the reviewer about the description of the chemistry being measured and we tried to be careful to refer to the DPPs as measuring “deacylation” while the APTs are doing “depalmitoylation”. We also added additional discussion about this point based on other reviewer’s comments. We also renamed the table. Thank you for this correction.

REVIEWERS' COMMENTS:

Reviewer #1 (Remarks to the Author):

The authors have done an excellent job addressing my critiques. I am pleased to recommend publication in Nat. Comm.

Reviewer #2 (Remarks to the Author):

The authors have provided a reasonably strong revision that has addressed most of my comments/critiques. My comments are below:

1. The authors address the mitochondrial localization of the APT1 enzyme using fractionation. This makes their *in vivo* studies with mitoDPP2 much stronger. Also, including the catalytically dead version of APT1 makes a strong argument for its localization and the localization of the mitoDPP2 probe. I think this alone (Figure 4) made the paper much better.
2. They show the quantification for their knockdowns and a few western blots. As expected, their knockdown efficiency isn't great for some genes (Supp Table 3), and they should show a bar graph, not a table, with the KD efficiently compared to WT. The table that they use now is not easy to interpret as they use fold changes rather than relative abundance.
3. Figure 10 legend, watch out for ATP2 vs APT2.
4. The order of the manuscript could be re-ordered so that Figure 4 (mito localization of APT1) comes before the *in vivo* mitoDPP2 activity assays.
5. They now have a HAP1 KO of APT1 but they only do localization experiments with it. Seems like it would be easy to do the mitoDPP2/3 experiments with the KO cell line.

Reviewer #3 (Remarks to the Author):

A number of important improvements have been made to the manuscript and many of the technical shortcomings have been resolved. The manuscript now clearly demonstrates mito-DPP2 hydrolysis in mitochondria. In the newly added figure 4, the authors perform a head-to-head comparison between DPP2 and mito-DPP2. This additional experiment demonstrates that mito-DPP2 is hydrolysed in the mitochondria and does not translocate and accumulate into the mitochondrial compartment after hydrolytic cleavage in the cytosol. Additionally, in supporting figure 2 it is now shown that APT1 hydrolyses mito-DPP2 at higher pH. The lack of monitoring the KD efficiency by RNAi has been resolved by control experiments on Western blot and qPCR. The data obtained with the plate reader assay with low statistical power has been backed with flow cytometry. To demonstrate the presence of APT in the mitochondria, improved fractionation experiments have been performed that show that APT1 is predominantly present in mitochondria.

Based on the revision, there remain two points which the authors should address in the final published article; publication is recommended provided these points are addressed (by commenting and acknowledging, not by new experiments):

Quantification: In figure 4E there is a large difference in fluorescence in mito-DPP2 while the quantification in F only shows an increase from 100 to 200%. How suitable, then, is microscopy for quantification?

The evidence for the authors conclusions is solid, however from an objective perspective it should be recognised that hydrolysis of mito-DPP2 does not necessarily correspond with physiologically relevant depalmitoylation activity because a) APT1 can have multiple functions b) there might be depalmitoylases insensitive for mito-DPP2 and c) mito-DPP2 might be hydrolysed by proteins with functions other than depalmitoylase activity.

Reviewer #1 (Remarks to the Author):

The authors have done an excellent job addressing my critiques. I am pleased to recommend publication in Nat. Comm.

Thanks!

Reviewer #2 (Remarks to the Author):

The authors have provided a reasonably strong revision that has addressed most of my comments/critiques. My comments are below:

1. The authors address the mitochondrial localization of the APT1 enzyme using fractionation. This makes their in vivo studies with mitoDPP2 much stronger. Also, including the catalytically dead version of APT1 makes a strong argument for its localization and the localization of the mitoDPP2 probe. I think this alone (Figure 4) made the paper much better.

Thanks, we also really like this new experiment.

2. They show the quantification for their knockdowns and a few western blots. As expected, their knockdown efficiency isn't great for some genes (Supp Table 3), and they should show a bar graph, not a table, with the KD efficiently compared to WT. The table that they use now is not easy to interpret as they use fold changes rather than relative abundance.

We added the raw data (CT values) for each experiment.

3. Figure 10 legend, watch out for ATP2 vs APT2.

Fixed.

4. The order of the manuscript could be re-ordered so that Figure 4 (mito localization of APT1) comes before the in vivo mitoDPP2 activity assays.

We think it still makes more sense to confirm localization of the probe first by confocal microscopy (Fig. 3) and then the localized activity control (Fig. 4)

5. They now have a HAP1 KO of APT1 but they only do localization experiments with it. Seems like it would be easy to do the mitoDPP2/3 experiments with the KO cell line.

We agree this is an interesting experiment to try, which we will do at some point. However, I don't think it would add value to this current paper given we already have the shRNA and inhibitor data.

Reviewer #3 (Remarks to the Author):

A number of important improvements have been made to the manuscript and many of the technical shortcomings have been resolved. The manuscript now clearly demonstrates mito-DPP2 hydrolysis in mitochondria. In the newly added figure 4, the authors perform a head-to-head comparison between DPP2 and mito-DPP2. This additional experiment demonstrates that mito-DPP-2 is hydrolysed in the mitochondria and does not translocate and accumulate into the mitochondrial compartment after hydrolytic cleavage in the cytosol. Additionally, in supporting figure 2 it is now shown that APT1 hydrolyses mito-DPP2 at higher pH. The lack of monitoring the KD efficiency by RNAi has been resolved by control experiments on Western blot and qPCR. The data obtained with the plate reader assay with low statistical power has been backed with flow cytometry. To demonstrate the presence of APT in the mitochondria, improved fractionation experiments have been performed that show that APT1 is predominantly present in mitochondria.

Indeed, thank you.

Based on the revision, there remain two points which the authors should address in the final published article; publication is recommended provided these points are addressed (by commenting and acknowledging, not by new experiments):

Quantification: In figure 4E there is a large difference in fluorescence in mito-DPP2 while the quantification in F only shows an increase from 100 to 200%. How suitable, then, is microscopy for quantification?

The fold difference, which is what you are referring to, is not a relevant measure in fluorescence microscopy with turn-on probes. The fold is based on the background, which is arbitrary for imaging. Imagine if the bar graph were just chopped off at some arbitrary level, the error bars

would not change, but the fold difference would. This could be functionally done by bringing down the excitation source, or oppositely by turning it up. Ultimately, turn-on probes are very good at quantifying whether two samples are different, not how different they are (if you count that as quantification). The error bars and stats are what matter.

All of that said, we recently published a new set of probes in Chemical Science that have a ratiometric response, which do now allow you to much more precisely quantify reaction progress in live cells by imaging.

The evidence for the authors conclusions is solid, however from an objective perspective it should be recognised that hydrolysis of mito-DPP2 does not necessarily correspond with physiologically relevant depalmitoylation activity because a) APT1 can have multiple functions b) there might be depalmitoylases insensitive for mito-DPP2 and c) mito-DPP2 might be hydrolysed by proteins with functions other than depalmitoylase activity.

We added additional text to our “caveats” paragraph that was already present in the discussion specifically stating each of these well-received points. Thank you.